



**1 Rapid waxing and waning of Beringian ice sheet reconcile glacial climate records from**

**2 around North Pacific**

Zhongshi Zhang[1,2,3,4]*, Qing Yan[4]*, Ran Zhang[5]*, Florence Colleoni[6], Gilles Ramstein[7], Gaowen Dai[1],
Martin Jakobsson[8,9], Matt O'Regan[8,9], Stefan Liess[10,1], Denis-Didier Rousseau[11,12], Naiqing Wu[13,14],
Elizabeth J. Farmer[15], Camille Contoux[7], Chuncheng Guo[2], Ning Tan[13], Zhengtang Guo[13,14]
1. Department of Atmospheric Science, School of Environmental studies, China University of Geoscience, Wuhan, 430074, China
2. NORCE Norwegian Research Centre, Bjerknes Centre for Climate Research, 5007 Bergen, Norway
3. Center for Early Sapiens Behaviour, 5007 Bergen, Norway
4. Nansen-Zhu International Research Center, Institute of Atmospheric Physics, Chinese Academy of Sciences, 100029, Beijing,
China
5. Climate Change Research Center, Institute of Atmospheric Physics, Chinese Academy of Sciences, Beijing 100029, China
6. Istituto Nazionale di Oceanografia e Geofisica Sperimentale, OGS, 34010 Sgonico (TS), Italy
7. Laboratoire des Sciences du Climat et de l'Environnement, LSCE/IPSL, CEA-CNRS-UVSQ, Université Paris-Saclay, F-91191
Gif-sur-Yvette, France
8. Department of Geological Sciences, Stockholm University, 10691, Stockholm, Sweden
9. Bolin Centre for Climate Research, Stockholm University, 10691, Stockholm, Sweden
10. Department of Soil, Water, and Climate, University of Minnesota, Saint Paul, MN 55108, USA
11. Laboratoire de Meteorologie Dynamique (CNRS and Institute Pierre Simon Laplace, IPSL), Ecole Normale Superieure, Paris
Sciences & Lettres (PSL) Research University, 75005 Paris, France
12. Lamont-Doherty Earth Observatory, Columbia University, Palisades, NY 10964, USA
13. Key Laboratory of Cenozoic Geology and Environment, Institute of Geology and Geophysics, Chinese Academy of Sciences,
Beijing 100029, China
14. College of Earth Sciences, University of Chinese Academy of Sciences, Beijing 100049, China
15. E. Farmer Science Editing and Writing, Bergen, Norway
Correspondence: Zhongshi Zhang (zhongshi.zhang@cug.edu.cn), Qing Yan (yanqing@mail.iap.ac.cn), Ran
Zhang (zhangran@mail.iap.ac.cn)

## 29 Abstract

Throughout the Pleistocene the Earth has experienced pronounced glacial-interglacial cycles, which have
been debated for decades. One concept widely held is that during most glacials only the Laurentide-Eurasian
ice sheets across North America and Northwest Eurasia became expansive, while Northeast Siberia-Beringia
remained ice-sheet-free. However, the recognition of glacial landforms and deposits on Northeast Siberia-
Beringia and off the Siberian continental shelf is beginning to call into question this paradigm. Here, we
combine climate and ice sheet modelling with well-dated paleoclimate records from the mid-to-high latitude
North Pacific to demonstrate the episodic occurrences of an ice sheet across Northeast Siberia-Beringia. Our
simulations first show that the paleoclimate records are irreconcilable with the established paradigm of
Laurentide-Eurasia-only ice sheets, and then reveal that a Beringian ice sheet over Northeast Siberia-
Beringia causes feedbacks between atmosphere and ocean, the result of which better explains these climate
records from around the North Pacific during the past four glacial-interglacial cycles. Our simulations
propose an alternative scenario for NH ice sheet evolution, which involves the rapid waxing and waning of



the Beringian ice sheet alongside the growth of the Laurentide-Eurasian ice sheets. The new scenario settles
the long-standing discrepancies between the direct glacial evidence and the climate evolution from around
the mid-to-high latitude North Pacific. It depicts a high complexity in glacial climates and has important
implications for our understanding of the dispersal of prehistoric humans through Beringia into North
America.

## 1. Introduction

Today, one popular understanding of Northern Hemisphere (NH) glaciations is that, beside Greenland, only
expansive Laurentide-Eurasian ice sheets existed during past glacials (Abe-Ouchi et al., 2013; Kleman et al.,
2013) and Northeast (NE) Siberia-Beringia was ice-sheet-free (Gualtieri et al., 2005). This concept was
established decades ago, after compelling evidence for an ice-free Wrangel Island (Gualtieri et al., 2005)
excluded the possibility of an ice sheet forming over NE Siberia-Beringia during the Last Glacial Maximum
(LGM). The region's low precipitation levels during glacials were posited as the dominant factor for limiting
ice growth (Gualtieri et al., 2005). Looking through the lens of this paradigm, considerable progresses
(Krinner et al., 2006; Yanase and Abe-Ouchi, 2010; Ganopolski et al., 2010; Ullman et al., 2014; Peltier et
al., 2015; Liakka et al., 2016; Colleoni et al., 2016; Tulenko et al., 2020) have been made in terms of
interpreting past glacial climate evolution and abrupt glacial climate events over the last few decades.
Among them, different climate models  (Yanase and Abe-Ouchi, 2010; Ullman et al., 2014; Liakka et al.,
2016; Colleoni et al., 2016; Tulenko et al., 2020) reproduce a robust feature that the Laurentide-Eurasian ice
sheets lead to cyclonic low-level wind anomalies over the North Pacific and warming feedbacks over NE
Siberia-Beringia.

However, the concept of Laurentide-Eurasia-only ice sheets is still under debate, in particular prior to the
LGM. Among the many points debated, the possibility of a pre-LGM ice sheet over NE Siberia-Beringia
suggested in many studies is key (Budd et al., 1998; Grosswald and Hughes., 2002; Bintanja et al., 2002;
Ziemen et al., 2014; Colleoni et al., 2016; Batchelor et al., 2019). A comparison between estimations of
Laurentide-Eurasian ice sheet volume and direct observations of sea level change during the LGM reveals a
discrepancy of unexplained missing ice with a volume of ~6-25 m ice-equivalent sea-level change (Simms
et al., 2019). Nevertheless, considerable glacial evidence is found across NE Siberia-Beringia, including
glacial sediments across Alaska (Darrell et al., 2011; Kaufman et al., 2011; Tulenko et al., 2018) and NE
Siberia (Stauch and Gualtieri, 2008; Glushkova, 2011; Barr and Clark, 2012a, b; Barr and Solomina, 2014),
marine deposits (Gualtieri et al., 2005) from ~70 ka in Marine Isotope Stage (MIS) 4 on Wrangel Island and
two glacial cirques (Stauch and Gualtieri, 2008) in the central part of the island, the orientation of



glaciogenic features mapped off the NE Siberian continental shelf (Niessen et al., 2013), a glacially scoured
trough on the outer margin north of De Long Islands (O'Regan et al., 2017), and glacial deposits on the New
Siberian Islands (Nikolskiy et al., 2017). Partly due to poor age controls, it remains highly controversial
(Barr and Clark, 2012a) whether the glacial evidence points towards a pre-LGM ice sheet over NE Siberia-
Beringia or local activities of ice domes/sheets on continental shelves (Niessen et al., 2013; O'Regan et al.,
2017) and mountain glaciers (Stauch and Gualtieri, 2008; Glushkova, 2011; Tulenko et al., 2018) on
continents.

The concept of Laurentide-Eurasia-only ice sheets should be compatible with the full range of paleoclimate
evidence if it is right, but a mounting number of paleoclimate reconstructions from around the North Pacific
show conflicts to the concept (e.g., Meyer and Barr, 2017; Bakker et al., 2020). Therefore, in this study, we
first review paleoclimate evidence from around the North Pacific, and then use climate and ice sheet
modelling to investigate whether the conflicts can be solved with the Laurentide-Eurasia-only ice sheets, and
whether an ice sheet over NE Siberia-Beringia is needed to reconcile the paleoclimate evidence.

This paper is organized as follows. Section 2 reviews paleoclimate records from around the North Pacific.
Section 3 describes our models and experimental designs. Section 4 and 5 show our results and discussions.
Section 6 is the summary.

**2. Paleoclimate records from around the North Pacific**
The two important evidence that could indicate glacial climate over NE Siberia-Beringia is the terrigenous
biomarkers from the Kamchatka Peninsula (Meyer and Barr, 2017; Meyer et al., 2017) and the
sedimentological facies in Lake El'gygytgyn (Melles et al., 2007, 2012). The biomarkers indicate the
summer surface air temperature (SAT) across NE Siberia during MIS 3/2 was almost as warm as today,
when NH summer insolation (NHSI) and atmospheric greenhouse gas levels were low. In Lake El'gygytgyn,
the cold sedimentological facies are characterized by laminations, high total organic carbon (TOC), total
nitrogen (TN), total sulphur (TS), and very low d13C$_{TOC}$ values. They were interpolated as permanent ice
covers due to extreme cold climates, which leads to ceased lake ventilations and anoxic bottom waters in
Lake El'gygytgyn.

However, it remains hardly to reconcile these two records (Melles et al., 2007, 2012; Meyer and Barr, 2017;
Meyer et al., 2017), as well as the evidence of ice advances over NE Siberia-Beringia (Stauch and Gualtieri,





2008; Glushkova, 2011; Kaufman et al., 2011; Barr and Clark, 2012a, b; Barr and Solomina, 2014; Tulenko
et al., 2018), in the concept of Laurentide-Eurasia-only ice sheets. Neither the feedbacks of sparse NE
Siberian vegetation due to the limited precipitation (Gualtieri et al., 2005), nor the feedbacks created by
enlarged Laurentide-Eurasian ice sheets (Meyer et al., 2017), yield a local summer warming across NE
Siberia. Furthermore, if the warming effect of large Laurentide-Eurasian ice sheets (Yanase and Abe-Ouchi,
2010; Ullman et al., 2014; Liakka et al., 2016; Colleoni et al., 2016; Tulenko et al., 2020) is explained as a
hamper (Tulenko et al., 2020) for ice accumulations and extreme cold climates over NE Siberia-Beringia,
the concept falls in conflict with the evidence of regional ice advances. If the climate effect of Laurentide-
Eurasia ice sheets (summer cooling (Meyer et al., 2017) as well as winter warming and increased moisture
supply (Meyer and Barr, 2017)) is explained as a favour (Tulenko et al., 2020) for the local ice
accumulations and advances, what mechanism limits the formation of an ice sheet over NE Siberia-
Beringia?

Comparing Devils Hole (DH, Nevada) $\delta^{18}$O (Landwehr et al., 2011; Moseley et al., 2016) and the NH ice
sheet evolution over the past four glacial-interglacial cycles provides crucial evidence to this debate. The
DH $\delta^{18}$O records show that towards the end of each of the last four full glacial cycles, the mean surface
temperature started increasing earlier in terrestrial regions on the mid-latitude North American west coast,
while the NH ice volume kept increasing (Fig. 1c). Such early warming also appears in the sea surface
temperature (SST) reconstructions at Ocean Drilling Program (ODP) Sites 1020 (Kreitz et al., 2000) and
1014 (Yamamoto et al., 2004) along the mid-latitude North American west coast. Indeed, with precise
Uranium-series age controls, the DH $\delta^{18}$O systematically shows delays of several thousand years between
the regional temperature and NHSI minimums (Fig. 1a). For example, during MIS 4, when NHSI reached its
minimum at 70 ka, the regional temperature remained high, with the temperature minimum actually
appearing four thousand years later at ~66 ka. During this isotope stage, the magnitude of the mid-latitude
cooling appears to be asymmetric around the North Pacific (Kreitz et al., 2000; Yamamoto et al., 2004;
Fujine et al., 2006; Rousseau et al., 2009; Landwehr et al., 2011; Moseley et al., 2016) — stronger in East
Asia than along the North Pacific eastern margin (Fig. 1d and Supplementary Fig. 1). A temperature
asymmetry occurred again ~40-30 ka  (Kreitz et al., 2000; Yamamoto et al., 2004; Harada et al., 2004;
Fujine et al., 2006; Harada et al., 2008; Rousseau et al., 2009; Landwehr et al., 2011; Moseley et al., 2016;
Maier et al., 2018), with a cooling in East Asia but a warming along the North Pacific eastern margin (Fig.
2). In the result section below, our simulations forced with the ICE6G ice sheet reconstructions (Peltier et
al., 2015) will investigate whether the growth of the Laurentide-Eurasian ice sheets alone can explain the
early warming and the asymmetry changes from around the North Pacific.



## 3. Modelling method

### 3.1 Introduction to models

The well-documented NorESM-L is a state-of-the-art earth system model (Zhang et al., 2012; Bentsen et al., 2013), developed at the Bjerknes Centre for Climate Research (BCCR), Norway. NorESM-L couples the spectral Community Atmosphere Model (CAM4) (Eaton, 2010; Neale et al., 2013) and the Miami Isopycnic Coordinate Ocean Model (MICOM). The resolution of spectral CAM4 is approximately 3.75° (T31) in the horizontal and 26 levels in the vertical. The resolution of the ocean is approximately 3° (g37) in the horizontal and 32 layers in the vertical. NorESM-L performs well in simulating the pre-industrial climate (Zhang et al., 2012) and has good skill in simulating paleoclimates (Zhang et al., 2013; 2014). CAM4 realistically simulates the NH trough-ridge system, in agreement with observations.

BIOME4 is an equilibrium biogeography model (Kaplan et al., 2003), widely used in simulations of equilibrated vegetation in past and future climate projections (Salzmann et al., 2009; Contoux et al., 2013). It simulates 28 biomes on a horizontal resolution of 0.5° latitude by 0.5° longitude, and uses the different bioclimatic limits (temperature resistance, moisture requirement and sunshine amount) among plant functional types to simulate the potential natural vegetation of a given climate.

The Parallel Ice Sheet Model (PISM) is a widely used (Golledge et al., 2015; Aitken et al., 2016; Yan et al., 2016; Bakker et al., 2017) three-dimensional, thermodynamically coupled continental-scale hybrid ice sheet model (Winkelmann et al., 2011; Martin et al., 2011; The PISM authors, 2015), run at a resolution of 40 km×40 km in this study. It uses the shallow ice approximation (SIA) and the shallow shelf approximation (SSA). Ice velocity is the sum of the velocities from the SIA and the SSA, which provides a consistent treatment for different flow regimes in ice sheets, streams, and shelves. Surface mass balance is the difference between snowfall accumulation and surface melting. The snowfall is determined based on the partitioned total precipitation following an empirical relationship relating total precipitation and air temperature. Surface melting is estimated according to the positive degree-day scheme (PDD), and the melted snow is able to refreeze as superimposed ice. Here, we set the daily melt rate to 5 mm/d°C for ice (PDD_ice), and 2 mm/d°C for snow (PDD_snow), with a standard deviation of 2.5 °C for the daily cycle of surface air temperature (Temp_std). Ice velocities are modulated by means of enhancement factors set to 1 for flow treated with SIA (ENF_SIA), and 0.1 for flow treated with SSA (ENF_SSA). Calving is solved based on eigen calving method (eigen_calving_K=$2\times10^{18}$ m/s) (Levermann et al., 2012). In addition, calving is triggered when the ice shelf front reaches 200 m (thickness_calving_threshold). Basal sliding is based on a pseudo-plastic power law model (Greve and Blatter, 2009) in which the exponent q is set to 0.25 (pseudo_plastic_q). These parameters are tuned in our equilibrated LGM experiments to produce favourable

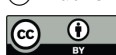



conditions for ice sheet growth (thus called FAV parameters), in which the simulated total NH ice volume
exceeds 100 m ice-equivalent sea-level change. They are used to simulate ice sheets over full glacial-
interglacial cycles. PISM includes a parameterization (Holland and Jenkins, 1999) to calculate sub-shelf
melt rates. In addition to the FAV parameters, we choose another set of PISM parameters to repeat the
experimental flow for full glacials, in order to consider uncertainties in ice sheet modelling. These
parameters (called IDL parameters) are tuned in the equilibrated LGM experiments to produce a Laurentide
ice sheet close to reconstructions and make the simulated NH total ice volume reached ~130 m ice-
equivalent sea-level change. We set PDD_ice to 2 mm/d°C, PDD_snow to 1 mm/d°C, Temp_std to 1 °C,
ENF_SIA to 1, ENF_SSA to 0.1, thickness_calving_threshold to 500 m, pseudo_plastic_q to 0.75, and
eigen_calving_K to $2 \times 10^{17}$ m/s.

## 3.2 Experimental design for NorESM-ICE6G simulations

To investigate climate responses to the Laurentide-Eurasian ice sheets, we use NorESM-L to carry out
multiple snapshot experiments and prescribe the ICE6G ice sheet reconstructions (Peltier et al., 2015) for
selected time slices during the last glacial-interglacial cycle. We select 21 time slices from 126 to 10 ka,
according to the relative maximums, minimums and mid-points of July insolation at 65 $^{\circ}$N (Berger and
Loutre, 1991). For example, to simulate the climate of 22 ka (called ICE6G-22ka), we prescribe global
modern vegetation cover and the ICE6G ice sheet extent (area and topography) of 22 ka, and set orbital
parameters and atmospheric greenhouse gas ($CO_2$ and $CH_4$) levels to their values at 22 ka. This simulation is
initialized from the previously simulated climate at 27 ka and run for 500 model years. The mean climate
from the last 100 model years is used to compare with the DH $\delta^{18}$O.

To further investigate the climate sensitivity due to the Laurentide-Eurasian ice sheets, we select two time
slices (22 and 70 ka) to provide two reference experiments. Only orbital parameters and atmospheric
greenhouse gas levels are modified to use their values of 22 and 70 ka. Vegetation cover, topography and ice
sheet distributions (Greenland and Antarctica only) are fixed and use modern conditions. The comparison
between the ICE6G-22ka (ICE6G-70ka) and the reference-22ka (reference-70ka) experiment can illustrate
the climate sensitivity due to the large-size (middle-size) Laurentide-Eurasian ice sheets.

## 3.3 Experimental design for NorESM-BIOME4-PISM flows

We design four experimental flows (Fig. 3) to simulate NH ice sheet variations during the past four glacial-
interglacial cycles. We use the same method to select time slices as used in the NorESM-ICE6G simulations.
Here, we use the last glacial-interglacial cycle (with 21 time slices) as an example to introduce the flow. At



the beginning, we use NorESM-L, forced with the 126 ka orbital parameters and atmospheric greenhouse
gas ($CO_2$ and $CH_4$) levels, and modern ice sheet distributions (Greenland and Antarctica only), to carry out a
climate simulation, and use this simulated climate to force PISM to get the NH ice sheets in equilibrium
with the simulated climate of 126 ka. Next, two iterations of climate simulations are run for 300 model years
each. In the first iteration, we prescribe the simulated 126 ka NH ice sheets to NorESM-L, with the orbital
parameters and atmospheric greenhouse gas levels set at 120 ka, to obtain a climate condition for 120 ka.
This climate is used to force BIOME4 to generate a vegetation cover in equilibrium with the simulated
climate at 120 ka. In the second climate iteration, the simulated vegetation (tundra and taiga in the NH high
latitudes) at 120 ka is prescribed in NorESM-L to simulate a new climate condition for 120 ka. This second
climate is prescribed in PISM to simulate the NH ice sheet extent at 120 ka. Note, at this step, PISM restarts
from the previously simulated 126 ka ice sheet extent and runs for 6 thousand model years only (between
126 and 120 ka). This experimental flow is repeated to simulate climate, vegetation and ice sheet extent for
each selected time slice. It allows us to carry out transient ice sheet simulations with time steps of 4-6
thousand years to mimic the NH ice sheet evolution during the past four glacial-interglacial cycles. In the
experimental flows, the SH ice sheet extent is fixed and uses the modern condition. Following the growth of
NH ice sheets, changes in seaways (the closing of the Bering Strait and the Barents Sea) are considered in
the experimental flows.

After we generate a scenario of ice sheet evolution, we carry out multiple climate snapshot experiments for
each time slice (K1-4 in Fig. 3). For example, we prescribe the simulated vegetation (tundra and taiga) and
ice sheet extent of 114 ka, together with the orbital parameters and atmospheric greenhouse gas levels of 114
ka, to generate the climate of 114 ka. This simulation is initialized from the climate of 120 ka and run for
500 model years.  We use the last 100 model years of these snapshot experiments to figure out the climate
evolution under the new ice sheet scenario during the past four glacial-interglacial cycles.

**4. Results**
**4.1 Can the Laurentide-Eurasia-only ice sheets reconcile the paleoclimate from around the North**
**Pacific?**
In consistent with previous studies performed with other models (Yanase and Abe-Ouchi, 2010; Ullman et
al., 2014; Colleoni et al., 2016; Liakka et al., 2016; Tulenko et al., 2020) , the NorESM-ICE6G experiments
show that the growth of the Laurentide-Eurasian ice sheets alters atmospheric circulation patterns, leading to
cyclonic low-level wind anomalies over the North Pacific and a strong warming over NE Siberia-Beringia
(Fig.4a-d and Supplementary Fig. 2). Consistent with the recent study (Tulenko et al., 2020), our simulations



also show a continent-wide Laurentide ice sheet (22 ka) causes a stronger warming over NE Siberia-
Beringia than an incomplete Laurentide ice sheet (70 ka).

However, in the mid-latitudes, our simulations show that anomalous low-level westerlies enhance Ekman
pumping and upwelling, reducing the surface temperature, both in ocean and over land, along the mid-
latitude North American west coast (the rectangle in Fig.4a-d). Coolings appear on both margins of the mid-
latitude North Pacific, not a warming on the eastern and a cooling on the western margin (Fig. 2). This result
is different to early simulations carried out with a slab ocean (e.g., Liakka et al., 2016, Tulenko et al., 2020).
The early simulations (e.g., Tulenko et al., 2020) forced with the Laurentide-Eurasia-only ice sheet can
produce the asymmetry temperature changes (a warming on the eastern and a cooling on the western
margin) in the middle latitude Pacific, but provide inconsistent temperature responses (warming) in ocean
and (cooling) on land along the mid-latitude North American west coast. Note such inconsistent land-sea
temperature responses are not supported by the paleoclimate records from DH and ODP Sites 1020 and
1014. Our experiments with a dynamic ocean component in NorESM-L capture the ocean feedbacks that
were missing in the early simulations (e.g., Liakka et al., 2016, Tulenko et al., 2020), and provide more
realistic simulations along the mid-latitude North American west coast.

Forced by the Laurentide-Eurasian ice sheets alone, no early warming (Supplementary Fig. 3a) is simulated
on land (DH) and in ocean (ODP Sites 1020 and 1014) at the mid-latitude North American west coast in the
last glacial-interglacial cycle. The simulated regional surface air temperature (SAT) keeps decreasing in
MIS3 and 2 until the beginning of the NH deglaciation.

**4.2 Paleoclimate records reconciled by an ice sheet over NE Siberia-Beringia**
Once an ice sheet over NE Siberia-Beringia is included in our transient climate and ice sheet simulations,
these conflicts can be resolved. The presence of the ice sheet over NE Siberia-Beringia leads to ice-
vegetation-atmosphere-ocean feedbacks (Fig.4e-h, 5 and Supplementary Fig. 3,4), strengthening the trough-
ridge system in the NH mid-to-high latitudes and causing cyclonic low-level wind anomalies that shift
westward over the North Pacific. On the eastern side of the cyclonic wind anomalies, anomalous
southwesterlies and southeasterlies foster the advection of warm water and low-level atmospheric warming
along the North American west coast. On the western side, anomalous northwesterlies cool mid-latitude East
Asia. Forced by the ice sheet over NE Siberia-Beringia, the simulated asymmetric responses in surface
temperature match the geological evidence (Fig. 1 and 2) from both the North Pacific margins.






Our simulations further indicate the timing and the extent of the ice sheet over NE Siberia-Beringia (named
the Beringian ice sheet, BerIS) during the past four glacial-interglacial cycles. It grows when glacial
sedimentological facies appear in Lake El'gygytgyn (Fig. 6a). Although modelling transient ice sheet
evolution unavoidably includes uncertainties (see the discussion section), our simulations illustrate that the
ice sheet can cover most of Beringia. The simulated extent (Fig. 5, 7 and Supplementary Fig.5, 6) agrees
nicely with the mapped distribution of glacial landforms across NE Siberia-Beringia (Stauch and Gualtieri,
2008; Darrell et al., 2011; Kaufman et al., 2011; Glushkova, 2011; Barr and Clark, 2012a,b; Niessen et al.,
2013; Barr and Solomina, 2014; O'Regan et al., 2017; Nikolskiy et al., 2017; Tulenko et al., 2018; Batchelor
et al., 2019). The simulated ice volume accounts for ~10-25 m ice-equivalent sea-level change (~20-30% or
more of simulated NH ice volume, Supplementary Fig. 5), coinciding with the volume of the missing ice
during the last glacial (Simms et al., 2019). Critically, ~20-25% or more of the ice mass distributes on the
submarine Chukchi and East Siberian continental shelf. Outside the mountain regions of Chukotka and
Kamchatka, much of the Siberian mainland remains ice-free (Fig. 5 and 7).

Our simulations show that astronomical forcing is the first-order driver for the BerIS, and the associated ice-
vegetation-atmosphere-ocean feedbacks become more dominant when the BerIS grows large. Forced with
NHSI and atmospheric greenhouse gas levels, weak changes in atmospheric circulation favour ice sheet
expansion, starting from a circum-Arctic ice sheet configuration that includes the BerIS, while cooling due
to enhanced albedo by glacial tundra (Tarasov et al., 2013) on NE Siberia-Beringia promotes the waxing of
the BerIS (Supplementary Fig. 7). During NH full glacial periods, the ice-vegetation-atmosphere-ocean
feedbacks lead to the waning of the BerIS and the transformation to the Laurentide-Eurasia-only ice sheet
configuration. In consequence, the deglaciation starts earlier over NE Siberia-Beringia (Fig. 6a, b), while the
other NH ice sheets continue growing (Fig. 6c).

Our climate simulations involving the growth and collapse of the BerIS produce a good agreement between
the DH $\delta^{18}$O (as well as ODP 1020 and 1014 SST) and the simulated regional SAT over the North American
west coast. Consistent with these records (Landwehr et al., 2011; Moseley et al., 2016; Kreitz et al., 2000;
Yamamoto et al., 2004), the simulated regional SAT starts warming earlier than the NH deglaciation (Fig.
6d and Supplementary Fig. 3b). At the end of full NH glacials, the feedbacks associated with the BerIS
cause a regional warming that leads to an early $^{18}$O enrichment in precipitation and ground water in the
North American west coast regions, while the NH ice sheets are still expanding. The agreement indicates
that, in addition to the growth of the Laurentide-Eurasian ice sheets and increased freshwater release (Maier





et al., 2018), such a fluctuation in the extent of a substantial BerIS is necessary to reconcile the paleoclimate
records from the mid-latitude North Pacific margins. The reconciliation cannot be achieved through the
growth of ice domes on the NE Siberian continental shelf or mountain glaciers on the NE Siberian continent
(see the discussion section and Supplementary Fig. 4, 6), since the small-scale glaciations across NE Siberia-
Beringia cannot cause strong climate feedbacks to match the paleoclimate records (Fig. 2) from around the
North Pacific.

## 5 Discussion

In this study, we use four steps to address the debate of ice sheet development during past glacial-
interglacial cycles. First, we review the paleoclimate climate records from around the North Pacific. These
records illustrate that the early warming occurred (both on land and in ocean) along the mid-latitude North
American west coast (Fig. 1), and the asymmetric changes in surface temperature appeared on both side of
the mid-latitude North Pacific during some glacials (Fig. 1 and 2). Second, we validate the climate model,
NorESM-L, and show it realistically simulates the climate responses caused by the Laurentide-Eurasian ice
sheets (Fig. 4a-d), in agreement with earlier studies. Third, we use the NorESM-ICE6G experimental flow to
indicate that the Laurentide-Eurasian ice sheets alone cannot explain these paleoclimate records from around
the North Pacific (Supplementary Fig. 2, 3). Finally, we use the NorESM-BIOME4-PISM experimental
flows to reveal that these climate records and glacial evidence across NE Siberia-Beringia are well
reconciled (Fig. 4-6), when the fast waxing and waning of the BerIS are involved. The simulated BerIS (Fig.
7) agrees reasonably with the direct glacial evidence across NE Siberia-Beringia.

Here, the modelling method is fully appropriate to the question tackled in this study. However, some
unavoidable modelling uncertainties should be further considered, since they are important and can be
improved and further addressed in future studies.

### 5.1 Vegetation feedback for the inception of BerIS

To quantify the impact of vegetation, we repeat simulations for the two time slices, 190 and 114 ka, but
forced with the modern vegetation (taiga or cold deciduous forest and tundra) (Tarasov et al., 2013)
conditions on NE Siberia-Beringia. The newly simulated climate is used to force PISM with the two sets
(FAV and IDL) of PISM parameters. All these ice sheet simulations restart from previously simulated ice
sheet geometry at 196 and 120 ka and run for 6000 years.





Our climate and ice sheet simulations demonstrate that the vegetation-albedo feedback is critical for the
inception of the BerIS (Supplementary Fig. 7), in addition to changes in NHSI and atmospheric greenhouse
gas levels. For example, at 190 and 114 ka, when the modern vegetation condition is prescribed, the
simulated climate cannot generate an ice sheet over the NE Siberian-Beringian continents, no matter which
set of PISM parameters is used (Supplementary Fig. 7a, b, d, e). When NE Siberia is covered by forests,
snow cannot accumulate over the area. The local albedo remains quite dark and local surface temperature
inhibits the growth of an ice sheet. On the contrary, when the area is mostly covered by tundra, as suggested
by the pollen records from Lake El'gygytgyn that show the gradual switch in vegetation from forest to
tundra (Tarasov et al., 2013), strong cooling due to the vegetation-albedo feedback allows a large ice sheet
to be formed over NE Siberia-Beringia (Supplementary Fig. 6c, f). Therefore, in our simulations the
inception of the BerIS is not caused by cold model biases in NE Siberia-Beringia, or uncertainties in our
modelling method or parameters.

**5.2 Uncertainties in ice sheet modelling**
Due to simplifications in schemes or parameterizations used in models, simulating transient ice sheet
evolution is a difficult task and unavoidably includes uncertainties. Many of these schemes and
parameterizations are widely used, for example, the positive degree-day scheme (PDD), but parameters used
always have a large range. Moreover, choosing one set of parameters for the whole NH clearly simplifies the
consideration of regional differences in ice sheet growth. Dust feedbacks (Ganopolski et al., 2010; Krinner
et al., 2006) are not considered in our simulations.

In our study, the simulated magnitude of the BerIS fluctuation relies on the PISM parameters. The FAV
parameters (see the method section) allow the BerIS to wax and wane realistically (Supplementary Fig. 8a,
b) but cause clear biases in the simulated size of the Laurentide ice sheet (Supplementary Fig. 8c) and the
simulated maximum NH ice volume (Supplementary Fig. 8d). Although the IDL parameters (see the method
section) can reduce the biases in the simulated maximum NH ice volume, the simulated fluctuation of the
BerIS becomes unrealistic. The IDL parameters make ice accumulated fast, but limit ice melting over NE
Siberia-Beringia.

The biases seem amplified in the full glacials. Here, we discuss the transient ice sheet simulations forced
with the FAV parameters. Compared to the estimations of glacial global sea level changes (Spratt and
Lisiecki, 2016), the simulated NH ice volume is reasonably good in the early glacials, but largely
underestimated in the full glacials (Supplementary Fig. 8d). For example, in the last glacial-interglacial
cycle, the simulated NH ice volume equals to ~43 m and ~59 m ice-equivalent sea level drops in MIS5d and



MIS4 (Supplementary Fig. 8d, 6j, 6k), which is consistent with the sea level reconstructions (Spratt and Lisiecki, 2016). However, in MIS3 and MIS2, the simulated NH ice volume remains ~50 m and ~60 m ice-equivalent sea level drops, only reaching the minimum MIS3 estimations and the middle of the MIS2 reconstructions. Moreover, there is a delay in the simulated ice growth in the MIS3/2 transition. Although the sea level (Spratt and Lisiecki, 2016) and climate (Fig. 2) records suggest the ice growth in MIS3/2 transition is rapid and the BerIS should reach its maximum size around ~30-40 ka, our simulations show the NH ice volume and the BerIS reach maximums in the simulation of 27 ka (Fig 6 and Supplementary Fig. 8). One reason for this delay is the coarse time steps used in our transient simulations. In MIS2, due to the simulated small Laurentide ice sheet that does not provide strong warming feedbacks, the simulated deglaciation is also delayed over NE Siberia-Beringia. In the simulation of 22 ka, the NE Siberian-Beringian continent is not ice free, still covered by ice, though simulated ice extent and thickness in the 22 ka experiment (Supplementary Fig. 8s) is much smaller than in the 27 ka experiment (Supplementary Fig. 8r).

Another weakness in our simulations is that the waning of the ice sheet on the NE Siberian continental shelf seems incorrect. For example, in MIS3 (Supplementary Fig. 8m-p), the ice sheet disappears over the NE Siberian-Beringian continent but remains on the continental shelf. This bias is caused by the fact that cold SST (instead of warm SAT) is used to control ice on the continental shelf in PISM simulations. Due to the coarse resolution of NorESM-L, few model grids can be changed to land, when the sea level is dropped. (The changes in major seaways, the Bering Strait and the Barents Sea, are considered in the climate simulations.) Thus, on the continental shelf, NorESM-L only provides cold SST (instead of warm SAT) for the ice sheet model, limiting ice melting on the NE Siberian continental shelf.

It should be noted that the simulated waxing of the ice sheet on the NE Siberian continental shelf remains reasonable, though the simulations cannot unequivocally resolve the ice sheet limits. For example, the simulated ice extent in MIS4 places substantially more ice near Wrangel Island than the larger BerIS simulated in MIS6e (Fig. 7). A slight reconfiguration of the ice sheet could leave Wrangel Island ice-free, while still allowing a substantial BerIS during MIS 4. Note the ice sheet on the NE Siberian-Beringian continental shelf is less important than the ice sheet on the continent in modifying the atmosphere and ocean circulations around the North Pacific (Supplementary Fig. 6).

Although the above uncertainties in the ice sheet modelling should be revisited in future studies to archive more realistic simulations for past ice sheet evolutions, these uncertainties do not influence the main logic in this study. Without the BerIS, even the reconstructed Laurentide-Eurasia-only ice sheets cannot reconcile





the glacial climate records from around the North Pacific (Fig. 1, 2). On the contrary, these records are well
explained when the BerIS is involved.

**5.3 Uncertainties in climate modelling**
The uncertainty that may potentially influence the main logic in this study comes from the model spread in
simulating glacial climate responses on the eastern margin of the North Pacific. Although almost all models
simulate the cyclonic low-level wind anomalies over the North Pacific — a robust feature — due to the
Laurentide-Eurasian ice sheets, the positions of the simulated cyclonic anomalies include a model spread
(Yanase and Abe-Ouchi, 2010). Some models simulate the anomalies close to the American continent (for
example this study), while others produce the anomalies further westward (for example the simulations
shown in Liakka et al., 2016; Tulenko et al., 2020). In the second model group, it remains possible to use the
Laurentide-Eurasia-only ice sheets to explain the early warming (Fig. 1) and the asymmetry changes (Fig. 1
and 2) from around the North Pacific, only when these models can produce consistence temperature
responses in ocean and on land in the middle latitude North American west coast. At the same time, a
mechanism must be found to reconcile the climate effect caused by the Laurentide-Eurasia-only ice sheets
and the evidence of ice advances over NE Siberia-Beringia during glacials. In future, a new benchmark
experiment for MIS 4 in Paleoclimate Modelling Intercomparsion Project (PMIP), will be valuable for
further constraining the model spread.

Another uncertainty, which should be considered in future studies but does not influence the main logic
here, is that the atmosphere component CAM4 underestimates the cooling over East Asia and the North
American east coast. As illustrated in the sensitivity diagnoses (Fig.4e-h and Supplementary Fig. 4), no
matter the size of the BerIS, the model simulates similar and small southward extension of the two troughs.
The simulated cooling sensitivity is less than 2 °C over mid-latitude East Asia and the North American east
coast (Supplementary Fig. 4), which is much smaller than the reconstructed 5-6 °C cooling in the Japan Sea
(Fig.1, Supplementary Fig. 1) (Fujine et al., 2006). The weak sensitivity clearly limits the Laurentide ice
sheet to grow large.

**5.4 A new scenario for NH ice sheet evolution**
Despite the uncertainties mentioned above, our simulations suggest that a more consistent picture appears
from within the glacial and paleoclimate evidence, when the rapid changes in the BerIS are considered. For
example, a large BerIS in MIS4, which accounts for ~1/3 of NH ice volume as suggested in our simulations,
can explain the relatively high surface temperature along the North American west coast (Kreitz et al., 2000;
Yamamoto et al., 2004; Landwehr et al., 2011; Moseley et al., 2016), the extensive ice expansion in Alaska



(Darrell et al., 2011; Kaufman et al., 2011; Tulenko et al., 2018) and NE Siberia (Stauch and Gualtieri,
2008; Glushkova, 2011; Barr and Clark, 2012a,b; Barr and Solomina, 2014), the poorly ventilated dark grey
to black finely laminated sediments in Lake El'gygytgyn (Melles et al., 2012), the raised marine deposits on
Wrangel Island (Gualtieri et al., 2005), the glaciogenic features off the NE Siberian continental shelf
(Niessen et al., 2013; O'Rgegan et al., 2017; Nikolskiy et al., 2017), and the coldest temperature during the
last glacial in mid-latitude East Asia (Fujine et al., 2006; Rousseau et al., 2009).

Therefore, we put forward a new scenario involving the waxing and waning of a BerIS, in conjunction with
the growth and decay of the Laurentide-Eurasian ice sheets, to explain past glacial climate evolution. There
are no conflicts in this scenario with an ice-sheet-free NE Siberia at the LGM, since based on our
simulations NE Siberia-Beringia was already deglaciated. Our simulations do argue that NE Siberia-
Beringia was glaciated just before the LGM, at ~40-30 ka, when the temperature asymmetry occurred across
the mid-latitude North Pacific margins (Fig. 2). The simulated ice sheet at this time interval extends across
Wrangel Island, where ice-margin sedimentological features are absent (Gualtieri et al., 2005; Stauch and
Gualtieri, 2008) and extensive glaciation is interpreted missing since MIS4 (Stauch and Gualtieri, 2008).
The discrepancy likely arises not only from the uncertainties in ice sheet modelling, but also from the
uncertainties in interpretations of glacial evidence due to the paucity of age controls in rock exposure and
organic sediments on the island (Stauch and Gualtieri, 2008). The ice sheet grows on the permafrost across
NE Siberian-Beringian terrestrial regions, but on the sediments and islands on the NE Siberian-Beringian
continental shelf. Different glacial processes could cause Wrangel Island ice free during this interval, while
enough ice accumulated across other NE Siberian-Beringian regions (Fig. 7). Note the ice sheet on the NE
Siberian land area is more crucial for modifying the atmospheric and ocean circulations around the North
Pacific (Supplementary Fig. 6). If our scenario is correct, a renewed assessment for the origin of raised
marine deposits (Gualtieri et al., 2005) and glaciogenic features (Stauch and Gualtieri, 2008) on Wrangel
Island is required, and the glacial sedimentological facies in Lake El'gygytgyn (Melles et al., 2012) and
other widely distributed glacial sediments on NE Siberia-Beringia (Stauch and Gualtieri, 2008; Darrell et al.,
2011; Glushkova, 2011; Kaufman et al., 2011; Barr and Clark, 2012a,b; Niessen et al., 2013; Barr and
Solomina, 2014; O'Rgegan et al., 2017; Nikolskiy et al., 2017; Tulenko et al., 2018) needs reinterpretation.

**6 Summary**
In summary, whether a pre-LGM BerIS once existed remains to be an open question. Based on the
understanding of glacial climate dynamics available now, it remains difficult for the concept of Laurentide-
Eurasia-only ice sheets reconciling many glacial climate records (Kreitz et al., 2000; Harada et al., 2004;



Yamamoto et al., 2004; Fujine et al., 2006; Harada et al., 2008; Rousseau et al., 2009; Landwehr et al.,
2011; Melles et al., 2012; Moseley et al., 2016; Meyer et al., 2017; Maier et al., 2018) from around the
North Pacific and glacial direct evidence (Stauch and Gualtieri, 2008; Darrell et al., 2011; Glushkova, 2011;
Kaufman et al., 2011; Barr and Clark, 2012a,b; Niessen et al., 2013; Barr and Solomina, 2014; O'Rgegan et
al., 2017; Nikolskiy et al., 2017; Tulenko et al., 2018) from NE Siberia-Beringia in one framework.

Our study urges that the possibility of a pre-LGM BerIS should not be neglected. Our simulations and
model-data comparisons suggest that the BerIS waxed and waned rapidly in the past four glacial-interglacial
cycles (i.e. last 425 ka) and accounted for ~10-25 m ice-equivalent sea-level change during its peak glacials.
The simulated BerIS agrees reasonably with the direct glacial  (Stauch and Gualtieri, 2008; Darrell et al.,
2011; Glushkova, 2011; Kaufman et al., 2011; Barr and Clark, 2012a,b; Niessen et al., 2013; Barr and
Solomina, 2014; O'Rgegan et al., 2017; Nikolskiy et al., 2017; Tulenko et al., 2018) and climate (Meyer and
Barr, 2017; Meyer et al., 2017; Melles et al., 2012) evidence from NE Siberia-Beringia, and reconciles the
paleoclimate records from around the North Pacific  (Kreitz et al., 2000; Harada et al., 2004; Yamamoto et
al., 2004; Fujine et al., 2006; Harada et al., 2008; Rousseau et al., 2009; Landwehr et al., 2011; Moseley et
al., 2016; Meyer et al., 2017; Maier et al., 2018). We propose that the pattern of past NH ice sheet evolution
is more complex than previously thought, in particular prior to the LGM. Moreover, the interval around 30
ka seems a critical time window for early human migration to North America (Goebel et al., 2008) through
transient terrestrial corridors available along the North Pacific coastal regions, when warm North Pacific
currents brought mild climates and abundant food and melting ice provided drinkable fresh water.

In near future, new field and marine field investigations across NE Siberia-Beringia, to acquire sea level
sequences, glaciostatic changes, and paleoclimate records in the Beringian regions, are clearly key targets to
provide more precise age controls and robust constraints to the extent and timing of the BerIS. With these
constraints, improvements in modelling ice sheet dynamics and associated climate feedbacks will revitalise
our understanding of changes in the glacial-interglacial cryosphere and climate evolution. To stimulate the
improvements, experiments of MIS 4 that further distinguish the climate feedbacks due to the BerIS and the
Laurentide-Eurasia-only ice sheets, could be a new benchmark in the PMIP. These multi-disciplinary model-
data comparisons are essential to provide a more reliable and realistic framework to prehistoric human
origins and global dispersal, as well as for our understanding of current and future climate changes.



**Fig. 1. Paleoclimate records from mid-latitude North Pacific eastern and western margins since 425 ka**. The Devils Hole (DH) δ¹⁸O from the North American west coast (black line for DH-11 and green line for DH2-D, Landwehr et al., 2011; Moseley et al., 2016) , and (a) the July insolation at 65°N (dark red line, Berger and Loutre, 1991), (b) the atmospheric $CO_2$ levels (light blue line, Luthi et al., 2008), (c) the LR04 global benthic δ¹⁸O stack (orange line, Lisiecki and Raymo, 2005), (d) the MD01-2408 alkenone SST from the Japan Sea (reddish brown line, Fujine et al., 2006), (e) the warm snail percentage from Luochuan, Chinese Loess Plateau (Rousseau et al., 2009). The crosses show age control points for the DH δ¹⁸O and the MD01-2408 SST. The shaded bars in (a) to (c) highlight the glacial stadials, when NHSI reached minimums while regional temperature in the North American west coast did not. The shaded bars in (d) highlight the intervals, when asymmetry changes appeared between North American west coast and East Asia. The snail fossil record has no absolute age controls, but is based on a correlated age model (Rousseau et al., 2009) between a loess magnetic susceptibility timescale based on magnetic reversals and the benthic δ¹⁸O stack. The grey (light green) bars in (e) indicate the warm snail percentage with total sample numbers smaller (larger) than 20. The geographical locations of these sites are illustrated in Fig. 5.

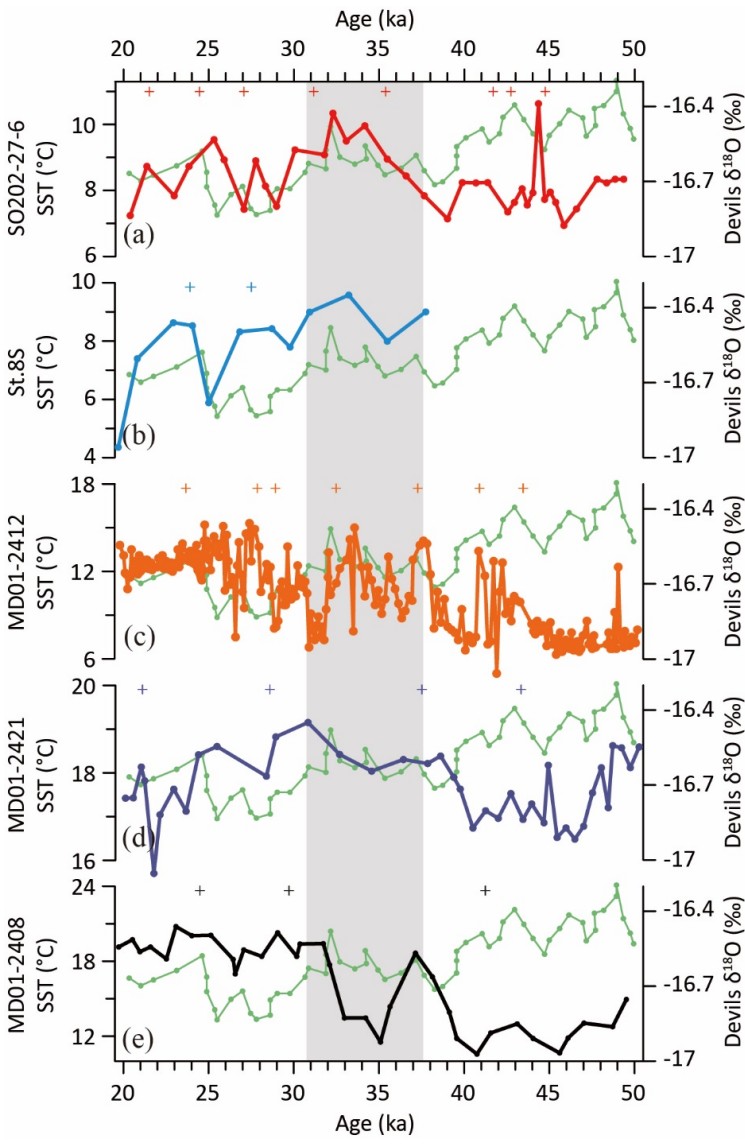

**Fig. 2. Devils Hole (DH) δ¹⁸O and alkenone SST from around North Pacific between 50 and 20 ka**. The DH δ¹⁸O (DH2-D, Moseley et al., 2016) and (a) the SO202-27-6 SST from the subarctic northeastern North Pacific (Maier et al., 2018), (b) the St.8S SST from the subarctic western North Pacific (Harada et al., 2004), (c) the MD01-2412 SST from the Okhotsk Sea (Harada et al., 2008), (d) the MD01-2421 SST from the Japan margin of the northwestern Pacific (Yamamoto et al., 2004), (e) the MD01-2408 SST from the Japan Sea (Fujine et al., 2006) (for the geographical locations, see map on Fig. 4). The crosses show ¹⁴C age controls on planktonic foraminiferal or age controls of tephra. The grey shaded bar highlights the interval with opposite SST changes between the eastern and western margins of the mid-latitude North Pacific. The interval is controlled by three ¹⁴C ages (31.30, 35.59, 41.82 cal ka) at the core SO202-27-6, two ¹⁴C ages (27.53, 52.79 cal ka) at the core St.8S, one tephra (Kc-1, 32.5 ka) and two ¹⁴C ages (27.85, 37.20 cal ka) at the core MD01-2412, one tephra (AT, 28.59 ka) and one ¹⁴C age (37.51 cal ka) at the core MD01-2421, one tephra (AT, 29.71 ka) and one ¹⁴C ages (41.23 cal ka) at the core MD01-2408. In the North Pacific, alkenone SST often indicates SST in main production period, in particular summer or autumn (Harada et al., 2004, 2008).



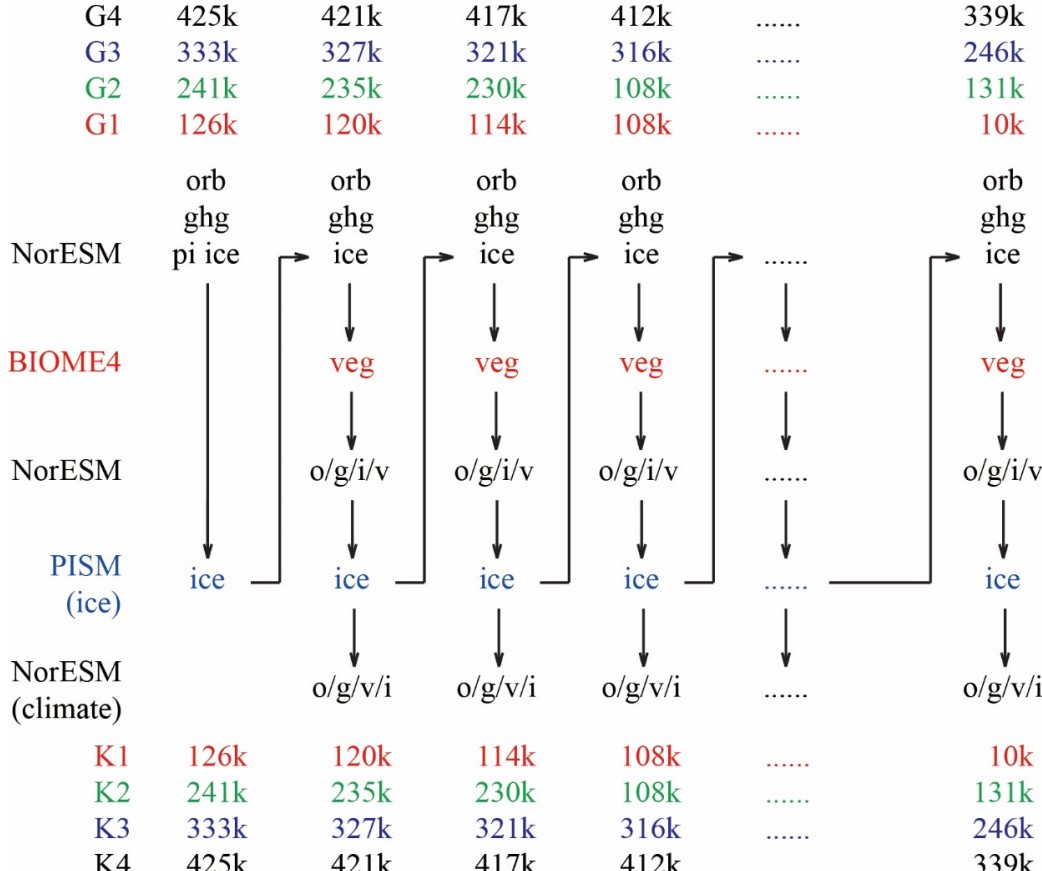

**Fig. 3. NorESM-BIOME4-PISM experimental flows.** The ice sheet outputs from PISM are used to illustrate ice sheet evolution during the past four glacial-interglacial cycles. The outputs from NorESM-L at the bottom line of the experimental flows are used to illustrate the climate evolution.

**Climate** Open Access
**of the Past**
**Discussions**

**Fig.4. Climate sensitivities in winter and summer due to ICE6G Laurentide-Eurasian ice sheets and simulated Beringian ice sheet.** Here, the climate sensitives mean the climate responses purely caused by the ice sheet imposed. Upper four panels, climate sensitivities in the 850 hPa winds (black arrows) and temperature (blue-brown shaded) due to the ICE6G ice sheet (Peltier et al., 2015) of 22 ka (a) and (b), and of 70 ka (c) and (d). Lower four panels, climate sensitivities due to the simulated BerIS of 190 ka in a large size (e) and (f), and of 114 ka in a small size (g) and (h). The climate sensitivities for other seasons are illustrated in Supplementary Fig. 2 and 4. The grey shaded areas show the distribution and height of ice sheets. The three red lines show the simulated 500 hPa geopotential heights in the ice sheet sensitivity experiments, while the three dashed blue lines show the results in the reference experiments. The black rectangle (between 35 and 45 °N, 115 and 135 °W) highlights the mid-latitude North American west coast, where DH, ODP Sites 1020 and 1014 are located.



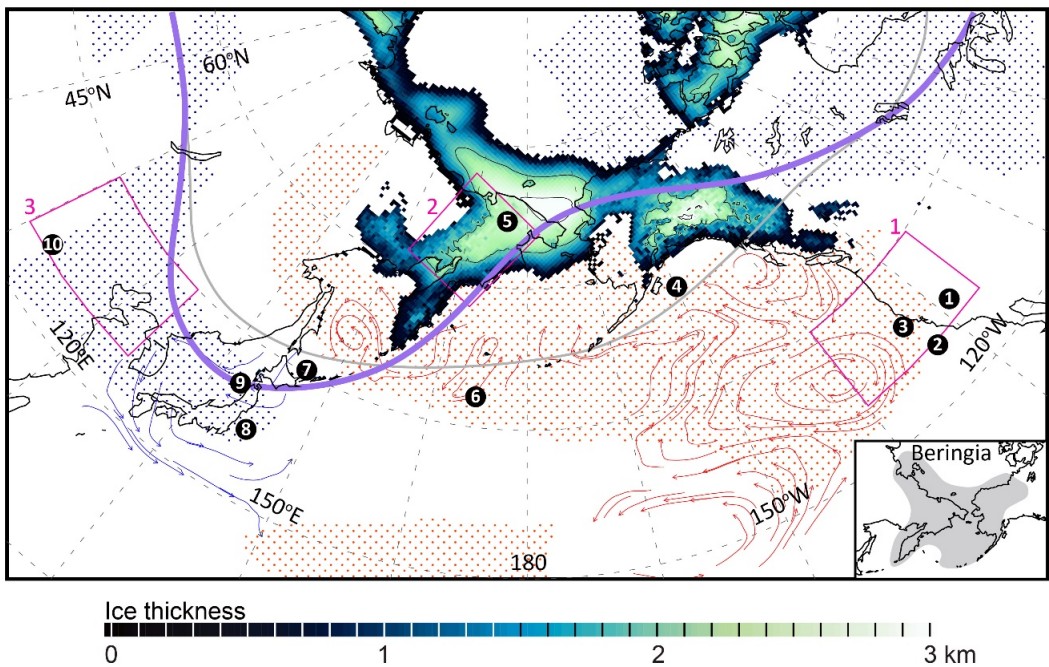

**Fig. 5. Schematic map of feedbacks caused by Beringian ice sheet**. The simulated Beringian ice sheet in
MIS4 is illustrated in the figure. The NH atmospheric stationary waves show a distinct trough-ridge
structure in the mid-to-high latitudes. Troughs lie over East Asia and the North American east coast, and a
ridge develops in between, lying over Beringia and the North Pacific. The purple line shows the deepened
trough (over East Asia and the North American east coast) and enhanced ridge (over the North Pacific and
Beringia) system, in comparison to the normal situation (the light grey line). As a result, cyclonic low-level
wind anomalies over the North Pacific intensify southwesterlies and southeasterlies over the North Pacific
eastern margin, which transport more warm ocean water (red lines and arrows) and heat northward to mid-
to-high latitudes. Warming (red dots) appears in the North American west coast, the North Pacific and the
Okhotsk Sea. Cooling (blue dots) appears in East Asia and the North American east coast, where the troughs
are deepened (also see Fig. 4 and Supplementary Fig. 4). The black markers show the geological data sites
used in the study. 1. Devils Hole (Landwehr et al., 2011; Moseley et al., 2016), 2. ODP Site 1014
(Yamamoto et al., 2004), 3. ODP Site 1020 (Kreitz et al., 2000), 4. core SO202-27-6 (Maier et al., 2018), 5.
Lake El'gygytgyn (Melles et al., 2012), 6. core St.8S (Harada et al., 2004), 7. core MD01-2412 (Harada et
al., 2008), 8. core MD01-2421 (Yamamoto et al., 2004), 9. core MD01-2408 (Fujine et al., 2006), 10.
Luochuan, Chinese Loess Plateau (Rousseau et al., 2009). Beringia (Hoffecker, 2007) includes the entire
stretch from the Mackenzie River in Canada to the Lena River in NE Siberia. The numbered boxes in this
figure are explained in the caption of Fig. 6.

**Fig. 6. Model-data comparison of ice sheet and climate evolution.** (a) The simulated ice thickness (purple shaded) averaged over NE Siberia-Beringia (the quadrilateral 2 marked in Fig. 5) compared to the July insolation at 65°N (red line) (Berger and Loutre, 1991). The glacial periods identified in Lake El'gygytgyn (Melles et al., 2012) are marked in bold and blue characters. The simulated BerIS grows in MIS 11b/a (~390 ka), 10c-a (~370-345 ka), 9d (~320-315 ka), 9b (~305 ka), 8c-a (~280-250 ka), 7d (~235-230 ka), 7b (~205 ka), 6e (~190-185 ka), 6c-a (~160-135 ka), 5d (~115 ka), 4 (~75-60 ka), 3/2 (~40-30 ka). These stages agree well with the glacial stadials identified in the Lake El'gygytgyn sediments, except for MIS 8a and 7b. (b) The simulated ice volume of BerIS (equals to sea level). The volumes larger than about twice of modern Greenland ice sheet are shaded in light blue. (c) The simulated total NH ice volume (equals to sea level, green shaded) compared to the LR04 global benthic $\delta^{18}O$ stack (orange line, Lisiecki and Raymo, 2005). (d) The simulated SAT (light magenta bars) averaged in the mid-latitude North American west coast (the box 1 marked in Fig. 5) compared to the DH $\delta^{18}O$ (black line for DH-11 and green line for DH2-D, Landwehr et al., 2011; Moseley et al., 2016). (e) The simulated SAT (orange bars) averaged in mid-latitude East Asia (the box 3 marked in Fig. 5) compared to the MD01-2408 SST in the Japan Sea (yellow line, Fujine et al., 2006) and the snail fossil record (blue line, Rousseau et al., 2009).



**Fig. 7. Simulated Beringian ice sheet in MIS6e (left) and MIS4 (right).** Upper panel (a) and (b) show the
ice thickness (km). Middle panel (c) and (d) show the surface mass balance (kg/m²y). Lower panel (e) and
(f) show the ice flow (m/y). The ice sheet extent is slightly larger in MIS6e than in MIS4. The simulated
extent agrees nicely with the distribution of glacial evidence across Alaska (Darrell et al., 2011; Kaufman et
al., 2011; Tulenko et al., 2018), NE Siberia (Stauch and Gualtieri, 2008; Glushkova, 2011; Barr and Clark,
2012a,b; Barr and Solomina, 2014) and the NE Siberian continental shelf (Niessen et al., 2013; O'Rgegan et
al., 2017; Nikolskiy et al., 2017). The Beringian ice sheet is also an important fresh water source that
perturbs glacial climate system, which should be investigated in future.





**Code availability**

NorESM is available from https://github.com/metno/noresm-dev.git. Documents of NorESM can be found on Geoscientific Model

Development: https://www.geosci-model-dev.net/special_issue20.html. BIOME4 can be downloaded from

https://pmip2.lsce.ipsl.fr/synth/biome4.shtml. PISM is available from https://pism-docs.org/.

**Data availability**

The paleoclimate data from around the North Pacific were previously published. The ICE6G ice sheet reconstructions are

available from the webpage: https://www.atmosp.physics.utoronto.ca/~peltier/data.php. All model outputs are available on

reasonable requests from the corresponding authors.

**Author contributions**

Z.Z. designed the study. Z.Z. carried out the NorESM-L simulations. Q.Y. carried out the PISM ice-sheet simulations. R.Z.

carried out the BIOME4 simulations. Z.Z., F.C and G.R. compared the current simulations with early studies. Z.Z, G.D., D.D.R,

E.J.F, M.J, M.O, Z.G. reviewed the glacial and paleoclimate evidence. Q.Y., R.Z., S.L, C.C., N.T. and C.G. contributed to the

diagnoses of model outputs. All authors contributed to discussion of the results and writing of the paper.

**Competing interests**

Author Denis-Didier Rousseau is a member of the editorial board of the journal.

**Acknowledgements**

The paper is dedicated to all pioneer scientists who were involved in the early debates of the Beringian ice sheet. All NorESM-L

simulations are carried on the cluster in the department of Atmospheric Science, China University of Geoscience.

**Financial support**

This study was jointly supported by the National Key Research and Development Program of China (Grant No.

2018YFA0605602), the National Natural Science Foundation of China (Grant No. 41888101, 41472160), the Norwegian

Research Council (Project No. 221712, 229819, and 262618), and the NordForsk-funded project GREENICE (Project No. 61841).

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
