# Peer review of "Rapid waxing and waning of Beringian ice sheet reconcile glacial climate records from 1"

_Climate of the Past, 2020_

## Referee Comment (RC1) · Anonymous Referee #1 · 12 May 2020

Review of "Rapid waxing and waning of Beringian ice sheet reconcile glacial climate records from around North Pacific"

This manuscript by Zhang et al. 2020 addresses an interesting and important issue, namely the spatiotemporal distribution of glacial ice-sheets in the northern hemisphere and the timing of presumed glacial evidence off the East Siberian Shelf. I thank the editor for the opportunity to review it, as well as the authors for this stimulating contribution to the discourse.

The analysis by Zhang et al. is novel and contributes to our understanding of a complex topic. I complement them on taking on a difficult task and gleaning useful insights from

their results. It is a good fit for Climate of the Past and I hope to see it published once they have addressed my concerns.

I recommend major revisions to this manuscript. Before publication, I would like to see a revised manuscript which focuses more on the mechanisms and feedbacks which can be studied from their experiments, and less on reconciling complex palaeoclimatic paradoxes which I do not think their experiments are particularly well-suited to address. I have provided feedback on both of these elements below.

I have a few main concerns that I would like the authors to address, followed by more specific recommendations based on line numbers below.

1) The main point of the paper seems to be that including the Beringian ice sheet in an asynchronously, two-way coupled climate/ice-sheet/vegetation modelling protocol "reconciles glacial climate records from around (the) North Pacific." However I do not see this conclusion clearly supported in the main figures. The most direct evidence to support this claim would be something like Supplementary Figure 3. However, I cannot tell by eye which of these two modelled scenarios (the orange and magenta bars) are more similar to the proxy records presented (black and green lines). No quantitative measures of how similar the timeseries are is presented (even as simple as cross-correlation of the records in question) to support the assertion that the orange bars provide a better fit to the data than the magenta bars. Furthermore this is difficult to assess because the y-axis limits for the two subplots are different. I would like to see this figure presented in the main text, with a full discussion of why subplot (b) indicates a better fit to the data than does subplot (a). In such a discussion, presenting the cross correlation (or other quantitative indices) to support the main claim of the paper is critical. Figure 1c is also very central to your argument – that the offset between Western NH surface temperatures and the deep-ocean d18O is not reconcilable with Laurentide-Eurasian only ice sheets. Please provide a figure that more directly compares your results with these proxy records to show how the inclusion of the Beringian ice sheet supports this interpretation. Suppl. Fig 3 has the last glacial cycle but why

not show the last 4 since you set up the paper with this in mind?

2) It is not immediately clear to me from the paper, figures, or supplement the extent to which the NorESM-BIOME4-PISM modelling approach actually reproduces realistic ice sheet extents, namely the extent of the LGM Laurentide ice sheet. This is central to the main argument of the paper, because I am concerned that the authors interpret changes in atmospheric circulation as being diagnostic of the presence of a Beringian ice sheet, when such features may equally be due to the lack of a proper Laurentide. I believe that Figure 4 is somewhat geared towards addressing this, where you show the results of the climate model for the same time slice with and without certain ice sheets, to demonstrate their impact alone. However, the argument about reconciling glacial climate records depends to large part on what happened during the LGM, and the ice sheet extents used to force the climate model in Figure4E-H are not representative of the last glacial period. Why not use the modelled extent of the Beringian extent within the region you define as Beringia, and then Ice6G outside of that, to isolate the effect of the Beringian ice sheet on its own? I understand that this is sort of what you are trying to get away from – this assumption that all glacial periods look like the LGM, which I agree is a bad assumption. But your analysis has also focused on identifying mechanisms that can be used to fingerprint the presence of the Beringian ice sheet and I feel that the mechanisms you present would be more credible if the modelling set up allowed for a realistic expression of the Laurentide ice sheet at LGM. This is relevant to, for example, lines 264–268. Are these changes expressed in the same way when a Laurentide is included? The mechanistic explanation of strengthening the trough-ridge system is true of the Laurentide as well (see Figure 4C&G); the magnitude of temperature change is less relevant, in my opinion, because the temperature along the US west coast is not solely a function of the atmospheric circulation over the Pacific but will also be influenced by the presence/absence of a large Laurentide Ice Sheet. I recognize that these kinds of modelling approaches are computationally intensive and you are intentionally not fixing the ice sheet positions in order to investigate the novel, previously unappreciated consequences of capturing a fluctuating Beringian ice sheet.

But statements like line 282-284 really distract from that overall project by making a direct comparison with the LGM, which your simulations are not well-suited to address because they do not reproduce a Laurentide ice sheet. Why not instead focus on the interesting elements which do not depend on being able to perfectly reconstruct the LGM ice sheets – for example, the importance of vegetation feedbacks 5.1 which has not received as much attention in the literature but which I find very compelling. You can also describe the dynamical consequences of the fluctuating Beringian ice sheet (e.g. Supplementary Figure 6) without saying that you propose this precise configuration for the last glacial cycle, but rather want to study what the impacts and effects of such an ice sheet are in general. To me any mention of the LGM in the context of your results is not very compelling, because your reconstructed LGM Northern Hemisphere ice cover does not compare favorably with what we know from physical evidence (e.g. Peltier et al. 2015).

3) Regarding the ice sheet model set-up, the "IDL" parameters produce much larger ice sheets during glacial periods than the "FAV" parameters. I would suggest you include a table of the parameter choices which could go in the supplement for interested readers. I was genuinely confused by "IDL" and "FAV" because I kept reading them as "ideal" and "favorite" which are basically the same thing. To me these seem like high mass balance and low mass balance end members, with "IDL" having less mass loss due to melting (lower PDD factors), and calving (both from thickness calving and eigen calving). Could you call them something else – high-MB and low-MB for example? (Anything else is fine too, just something that makes it easier to differentiate between them for the reader). And to be clear, the IDL parameters produce northern hemisphere ice cover with a sea level equivalent of ∼130m (Supplementary Figure 8E) but with an extent restricted to approximately what is shown in Supplementary Figure 5L?

I really appreciate that these authors have taken on such an interesting and rich topic, on which little modelling work has previously been done. However I think that the current framing "reconciles glacial climate records" is not supported by the figures/text.

I would advocate for describing and analyzing the simulations while being more forthcoming about what questions these simulations can, and cannot, directly address. This has been done to some extent in the "uncertainties" sections, but these to me seem very focused on small details rather than the big picture.

Some finer details below, organized by line number:

• Line 2 "the" North Pacific • Line 36 replace "demonstrate" with "to evaluate the climatic consequences of" or similar • Line 36-40 This is a very strong claim and I would like to see it better supported in the text and figures. • Line 95 The two important "pieces" of evidence • Line 101 interpolated replace with "Interpreted" • Line 105 hardly replace with "hard" • Line 180-181 The IDL parameters were tuned to the volume of the Laurentide, right? Rather than the extent? That's important to clarify here. Saying "close to reconstructions" to me implies that the measured extent of the Laurentide (e.g. Ice-6G) was used as a constraint. • Line 236 "Consistent with" • Line 274-275 I would be cautious with the assertion that your modelled Beringian ice sheet is really indicative of the last four glacial cycles. I think the mechanism is robust and its important to show there could be an ice sheet, but the precise configuration is probably not captured by your model (for example, it is my understanding that Lake El'gygytgyn has never had an ice sheet over it, as there has been continuous sedimentation throughout the last 3.6 Ma and even during glacial periods (e.g. Melles et al. 2012, Science)). How can this be reconciled with <2 kms of ice (your figure 5) over this site? In my opinion the strength of your approach is understanding the mechanisms and feedbacks related to a Beringian ice sheet, rather than in identifying its actual extent/thickness. • Lines 293-296 I do not see what you are talking about regarding the diachronous retreat of the Beringian vs. NH ice sheets. They seem to have max ice volume at the same time (Figure 6B&C). • Lines 303-304. This is really central to your argument.. please point to a figure here • Lines 449-451 This interpretation is not supported by Supplemental Figure6Q–S where a Beringian ice sheet is present • Lines 456–458 "The ice sheet grows..." check this sentence – it does not make

sense to me as written currently.

Figures In general I feel the figures could be simplified by reducing the number of times you plot the same timeseries. For example in Figure 1 the DH speleothem record is plotted 4 times and in Figure 2 the same record appears 5 times. I think this makes comparisons more difficult and really implies we should be seeing the similarity of the records rather than using some kind of quantitative metric (cross correlation for example) to see where the records share variance. Where possible, please have each record only appear once (I know that sometimes plotting the record again is warranted, so I am just asking that you take it as a general principle to only plot each record once per plot, and I recognize there will be exceptions.) Please note that the locations of all the cores are shown in Figure 10. In general I do not feel like the core names are particularly illuminating in the actual axis labels, I would prefer geographic labels (i.e. Northwest Pacific SST, or simplified LAT/LON), with the core names listed in the caption.)

Figure 4 – Temperature is really Change in Temperature relative to the simulation with no ice sheet, right? Why are some of the areas white? No change, change not significant at some threshold...? (For example I'm thinking of the white areas where you have placed the "5" on top of the arrow. By the way, what are the units for those arrows?)

Figure 6 – Why are these timeseries shaded in below what seems to be an arbitrary value? Does this value mean something – i.e. it's 0.5km in A, ∼12m in B, 15m in C, etc. Why not just plot these as a time series as well – what benefit is filling it in? Again insolation is plotted twice. The pink and orange "bars" are confusing here as well, why are the plotted from 13.25 degrees in D and 4 degrees in E? Why not just use the black line (the time series) and color it? Perhaps with dots along it where you have actually run the model (which is what I assume the bars are trying to show)? It's confusing the way it is currently. Please use (m) rather than (10m) for the unit of subplot C.

Figure 7 – This is a really interesting figure, thanks for showing these model outputs.

Supplement

Figure 1 – the same timeseries is shown 5 times. Why? "Percentage" should range from 0-100 Figure 3 – see my comments above – this figure should be in the main manuscript, extended to 4 glacial cycles, and with timeseries instead of the vertical bars for the model outputs. Please do some analysis to show that Fig 3b better agrees with the proxy records. Please make the y-axis of both simulation outputs the same so that it is clear the difference in magnitude of the changes. Can you scale the d18O to a change in temperature using published relationships, or discuss in the text how you think the magnitude of the modelled changes in temperature would be expressed in terms of the DH d18O record? Figure 4 – again, should be change in temperature I think. Figure 5 – Which set of ice sheet model parameters is used here? FAV? Please include. "North American east coast" should be "Hudson Bay" or some other geographically relevant description (Northern Canada, etc.) – "east coast" is confusing. Figure 6 – quite generous to say the ice sheet is "largely gone" at 22ka, especially given current thinking that this ice sheet did not exist at the LGM. I would revise the caption to remove this. Still interesting to see how the ice sheet fluctuates along with the climate – thanks for including this. Figure 8 – if you are going to fill all of these in fill them in above 0 – it's confusing that they seem arbitrarily filled in. But since we cannot see the "IDL" scenarios behind the "fav" scenarios I would plot them just as time series, or find a way (partial transparency?) to make both sets of simulations visible throughout to see how the simulations differ during the early glacials. In E, seems like you meant to just show the purple, so that you are comparing ice thicknesses for two different regions from the same set of simulations. Throughout, the unit (10m) is confusing and I would advocate for (m).

---

## Author Comment (AC1) · 13 May 2020

Dear Reviewer,

Thanks for the constructive suggestions and comments. Here, we reply your major concerns (point 1 and 2). We think these two related points form the basis for the review.

We agree with you that Fig. 1c is very central to our argument. There is an offset between the mid-latitude North American west coast and the deep-ocean d18O. This offset indicates that in the mid-latitude North American west coast warming (both in ocean and on land) starts earlier than the NH deglaciation. In the paper, we call it "early warming" in the mid-latitude North American west coast.

[Figure]

Figure 1c. The LR04 global benthic $\delta^{18}O$ stack (orange line, Lisiecki and Raymo, 2005) and the Devils Hole (DH) $\delta^{18}O$ from the North American west coast (black line for DH-11 and green line for DH2-D, Landwehr et al., 2011; Moseley et al., 2016)

Supplementary Figure 3 is the important evidence to support our argument. We use it to demonstrate which ice sheet scenario, the Laurentide-Eurasian only or the Beringian-involved, can generate the early warming. In this model-data comparison, the trend is the most important.

[Figure]

Supplementary Fig. 3. Simulated SAT evolution in mid-latitude North American west coast during last glacial-interglacial cycle.

When the reconstructed Laurentide-Eurasian only ice sheets are included, please note that the

simulated surface temperature (Supplementary Figure 3a) keeps decreasing from 60 ka to 22 ka. In other word, if only the Laurentide-Eurasian ice sheets exist, no early warming occurs in the mid-latitude North American west coast, which conflicts with the climate records. The mechanism behind is revealed in Fig.4a-d in the main texts.

On the contrary, when the simulated Beringian ice sheet is involved, the simulated surface temperature (Supplementary Figure 3b) in the mid-latitude North American west coast show an increasing trend from 40 ka, much earlier than the NH deglaciation. In other word, the Beringian ice sheet allows a successful explanation for the early warming in the mid-latitude North American. The mechanism behind this success is revealed in Fig.4e-h in the main texts.

The mode-data comparisons for the past four glacial-interglacial cycles are shown in Fig. 6d in the main texts. It shows that our experiments successfully simulate the early warming for each glacial-interglacial cycle.

[Figure]

Figure 6d. The simulated SAT (light magenta bars) averaged in the mid-latitude North American west coast (the box 1 marked in Fig. 5) compared to the DH $\delta^{18}$O (black line for DH-11 and green line for DH2-D, Landwehr et al., 2011; Moseley et al., 2016).

We admit that our simulations underestimate the size of Laurentide ice sheet. This is a weakness in the current study. You ask a good question whether the underestimation makes our simulated TS fit better with the records. Our answer is no. We agree that the size of Laurentide ice sheet will influence the absolute values for simulated temperature in the mid-latitude North American west coast, but not the trend. The cooling effect due to the growth of Laurentide ice sheet must be compensated with a warming effect, otherwise the early warming can not happen.

We agree that it is crucial to understand the interaction between the Laurentide and the Beringian ice sheet. It is a difficult question. We once tried to understand the interaction with equilibrated experiments. Please read our early paper "Instability of Northeast Siberian ice sheet during glacials" https://www.clim-past-discuss.net/cp-2018-79/. The basic mechanism behind this interaction is revealed in the paper. We are going to use high-resolution climate models, as well as increased time steps, to further improve our simulations. This is our future task but with a huge effort, which may need two or three more years.

Moreover, to answer this interaction question also needs modelling intercomparisons with different climate models to constrain modelling uncertainties. Unfortunately, the possibility of a pre-LGM Beringian ice sheet was often neglected and thought to be wrong in the paleoclimate community. We hope our current paper can push the community to reconsider the Beringian ice sheet and re-evaluate

the established Laurentide-Eurasian-only concept.

In the minor points, you ask the question how the sediments from Lake El'gygytgyn can be reconciled with an ice sheet over it, since the lake receives continuous sediments during the Quaternary. As we wrote in the main texts, "In Lake El'gygytgyn, the cold sedimentological facies are characterized by laminations, high total organic carbon (TOC), total nitrogen (TN), total sulphur (TS), and very low d13CTOC values. They were interpolated as permanent ice covers due to extreme cold climates, which leads to ceased lake ventilations and anoxic bottom waters in Lake El'gygytgyn." Lake El'gygytgyn could be a subglacial lake when there is an ice sheet on it. It remains possible to explain these sediments in cold periods in a subglacial lake environment. This possibility should be reassessed.

We hope our reply can convince you. If you have any more questions or comments, please let us know.

Best regards

Zhongshi on behalf of all co-authors

---

## Referee Comment (RC2) · Julie Brigham-Grette (Referee) · 18 May 2020

Review of CP doi.org/10.5194/cp-2020-38 Authors: Zhang et al. 2020

The paper by Zhang et al (submitted) is an important attempt to reconcile evidence from numerous authors for the existence of some type of ice sheet over the East Siberian Sea during past glacial intervals, prior to MIS2. The concept of a Beringian Ice Sheet was initiated by Grosswald and Hughes (e.g., QSR 2002) for the Last Glacial Maximum (LGM, 20ka) but the physical evidence for that ice sheet has been missing for the LGM (see recent community map compilation by Bond, 2019). Zhang et al. however regurgitate arguments from decades past while trying to approach a modern problem.

[Figure]

I suggest they restructure the argument.

I think we all can agree that there is, yes, compelling evidence for some type of grounded glacial ice in the past off the Chukchi Cap and Arliss Plateau in the Arctic Ocean based on papers by for example, Niessen et al. (2013) and Joe et al. (2020, and references there in). Basilyan et al. (2010) show glacial ice shoving onto the New Siberian Islands before 70ka and more likely during MIS 6. The challenge, however, has been to determine what that ice sheet looked like and when was it present (Brigham-Grette, 2013). This remains a challenge because stratigraphic records both on and offshore, well-dated glacial moraine sequences in Chukotka and Alaska and regional paleoshorelines demonstrate a rich Pleistocene history without continental-scale glaciation. This paper takes a completely wrong approach to the problem.

There is no doubt that glacial ice seems to have been grounded on the outer shelf of the East Siberian Sea during earlier glacial episodes. While mammoths roamed Wrangel Island from 48 ka to nearly 3400 years ago (Vartanyan et al., 1993) there are erratics on the islands plateaus that prove it was glaciated earlier than that (Gualtieri et al. 2003). But the ice sheet that produced the glacial erosional and depositional features observed does not need to look like the ice sheets proposed in this paper. Rather, the reconstructions from this paper violate the physical evidence from many dozens of field-based reconstructions published over the past 3-4 decades. Their disregard for the field evidence is negligent in trying to get to a real issue. For example, Figure 7 has a caption suggesting that their simulation of a Beringian Ice Sheet for MIS 4 and MIS 6 "agrees nicely" with field based work cited, but this is simply not true. Supplemental Figure 5 demonstrates that their simulation does not even get the Laurentide or Eurasian ice sheets correct (note here that in S.Fig 5i they show the southern limit of the ice sheet in MIS 6 terminating in the center of Hudson's Bay, labeled "The North American east coast" in the caption).

The authors are asking the right questions but do not have the right approach. I suggest the authors first demonstrate how their simulation gets the Laurentide, Cordilleran, and

Eurasian ice sheets correct for MIS 2; produces glacial ice extent for MIS-2 Beringia as outlined in the map by Bond (2019). There is wide spread evidence for aridity in much of Beringia during the LGM. So they could address the question why MIS 6 and especially MIS 8, 10 and 12 were wetter (more extensive valley glacier systems). Next start with a small ice sheet in the East Siberian Sea. They could ask "how large can the model grow an East Siberian ice sheet without violating the geologic record. Felzer (2001) tried this experiment with a GCM that looks primitive by today's standards perhaps, but the approach respected the Beringian Pleistocene history known at the time. While cold-based glacial ice can miraculously preserve ancient landscapes, ice sheets still have to weigh something, and leave a record of glacial isostasy (see arguments in Brigham-Grette and Gaulteri, 2004)

There are three solid reasons why this paper should be rejected, but the most obvious is the mismatch with physical data:

1. This paper suggests central Chukotka, home to Lake El'gygytgyn, was repeatedly covered by glacial ice sheets nearly 2 km thick during the last several glacial cycles. This idea is ludicrous but central to their faulty glacial reconstructions. It is clear from Melles et al. (2012) and Brigham-Grette et al. (2013) that the central part of Chukotka was never covered by continental scale ice sheets over the past 3.6 million years. Absurdly, Zhang et al. suggest that Lake El'gygytgyn has been repeatedly covered by 1.5 to 2 km of glacial ice during the times when Melles et al. 2012 and notably dozens of subsequent papers support the interpretation that Lake El'gygytgyn was covered by a perennial lake ice cover during glacial times, much like perennially ice-covered lakes in the Antarctic Dry Valleys today with moats in summer. The "glacial" facies outlined by Melles et al. (2012) and Brigham-Grette et al. (2013) describes lacustrine (not glacial) sediments deposited under the lake-ice cover (not an ice sheet), consistent with the presence of cold and arid pollen assemblages and 10% biogenic silica produced by diatoms living in the water column under the lake ice. The authors of this present paper have completely ignored the meaning of glacial /interglacial/super-interglacial

change in the El'gygytgyn record. This undermines the premise of their paper from the start.

2. This paper ignores decades of research that documents the extent of glacial ice cover in Beringia during the LGM about 22 ka ago, and for earlier glacial intervals MIS 4,6, 8 10 etc. If you look at their figures Supplement #6 and 7, it is clear that most of their reconstruction violates field evidence (Brigham-Grette and Gualtieri, 2004, provides background). See also Dyke et al. (2002).

3. The argument of using ICE6G does not follow on their figures showing model outputs. And the argument that Devils Hole must be reconciled with the ice sheet extent seems like a huge stretch. Why should a record from Nevada say anything about a Beringian Ice Sheet? What is the physical teleconnection? The so-called missing ice sheet issue accounts for 10 meters of sea level but this is well within the error of models for glacial isostatic loading (GIA) and dynamic topography. This part of the paper seems weakly argued. The statement that "early warming is a better match to the paleoclimate data" is wrong given that the glacial onset and reconstructions are incorrect.

References:

Basiliyan et al. 2010. Age of cover glaciation of the New Siberian Islands based on 230Th/U-dating of mollusk shells. Structure and Development of the Lithosphere. Moscow. Paulsen, pg. 506-514.

Bond, J.D. 2019. Paleodrainage map of Beringia. Yukon Geological Survey, Open File 2019-2. The Yukon Geological Survey recently produced an amazingly accurate reconstruction of Beringian glaciation based on input from dozens of experts who have documented their field work. See https://www.livescience.com/64786-beringia-map-during-ice-age.html

Brigham-Grette, J. 2013. A fresh look at Arctic ice sheets. Nature Geoscience 6, 807-808

Brigham-Grette, J., Melles M., Minyuk,P., Andreev, A, Tarasov, P., DeConto, R., Koenig, S., Nowaczyk, N., Wennrich, V., Rosén P., Haltia-Hovi, E., Cook, T., Gebhardt, T., Meyer-Jacob, C., Snyder, J., Herzschuh, U. 2013. Pliocene Warmth, Polar Amplification, and Stepped Pleistocene Cooling recorded in NE Arctic Russia. Science 340, 1421- 1427. DOI:10.1126/science.1233137; plus supplemental.

Brigham-Grette, J. and Gualtieri, L., 2004. Response to Grosswald and Hughes (2004), "Chlorine-36 and 14C chronology support a limited last glacial maximum across central Chukotka, Northeastern Siberia, and no Beringian ice sheet," and, Gualtieri et al. (2003), Pleistocene raised marine deposits on Wrangel Island, NE Siberia: implications for Arctic ice sheet history. Quaternary Research, v. 62 (2), 227-232.

Dyke, A.S., Andrews, J.T., Clark, P.U., England, J.H., Miller, G.H., Shaw, J., Veillette, J.J., 2002. The Laurentide and Innuitian ice sheets during the last glacial maximum. Quat. Sci. Rev. 21, 9-31. (some recent updates put ice over Banks Island and a few other minor changes)

Felzer, B., 2001. Climate impacts of a ice sheet in East Siberia during the Last Glacial Maximum. Quat. Sci. Rev 20, 437-447.

Grosswald, M.G., Hughes, T.J., 2002. The Russian component of an Arctic ice sheet during the Last Glacial Maximum. Quaternary Science Reviews 21, 121– 146.

Joe, Y.J., Polyak, L., Schreck, M., Niessen, F., Yoon, S.H., Kong, G.S. and Nam, S-Il. 2020. Late Quaternary depositional and glacial history of the Arliss Plateau off the East Siberian margin in the Western Arctic Ocean. Quaternary Science Reviews. 228, 106099.

Melles, M., Brigham-Grette, J., Minyuk, P., and others. 2012. 2.8 Million Years of Arctic Climate Change from Lake El'gygytgyn, NE Russia. Science 337, 315-320, plus supplement.

Niessen, F, and 10 others. 2013. Repeated Pleistocene glaciation of the East Siberian
continental margin. Nature Geoscience 6, 842-846.
* * *

---

## Author Comment (AC2) · 23 May 2020

Dear Julie,

Thanks for the review. Although we are not surprised that you suggest rejecting our paper, we are happy to have a chance to discuss with you. We would like to use this chance to show our respects to your important contributions, the sediments from Lake El'gygytgyn, in understanding past climate evolution over Beringia.

Please allow us to ask you three questions.

1) How to resolve the conflicts in the perennial lake-ice explanation? Let us take two

stages, MIS3b, MIS10c. The Lake El'gygytgyn records show different sedimentological facies in these two stages, glacial facies in MIS10c, but interglacial facies in MIS3b. However, summer insolation, greenhouse gas levels, and the total NH ice volume (indicated from the benthic d18O) are very similar in these stages, indicating similar surface temperatures over NE Siberia-Beringia. With similar climate conditions, why did the glacial facies appear in MIS10c, but the interglacial facies in MIS3b. This conflict should be resolved if the lake-ice explanation is reasonable. The lake-ice explanation highly depends on the assumption that no gaps exist in these sediments in glacial facies. However, this is not the case, bringing uncertainties in the lake-ice explanation. (Please also see our reply below.)

We appreciate that you admit that ice sheet can develop on the East Siberian continental shelf, at least during some stages, for example MIS6. You also suggest that a wetter condition in MIS6 (than in LGM) is the key for understanding the larger ice sheet during MIS6. This suggestion already includes one conflict. Very similar external climate forcing should cause similar precipitation in MIS6 and LGM over Beringia. Our second question is related to this wetter condition explanation.

2) How to resolve the conflicts between MIS6 and MIS4? MIS6 and MIS4 have similar low summer insolation, while the greenhouse gas levels are higher in MIS4. These climate forcings lead to a wetter condition over Beringia in MIS4 than MIS6. Did the ice sheet develop over the continental shelf in MIS4? If the answer is no, it means precipitation is not the major controller for the ice sheet growth over NE Siberia-Beringia. If the answer is yes, to some extent it falls in conflict with the ice-free Wrangel Island.

The following conclusion is taken from a recent study (Bakker et al., 2020). "In most of the examined PMIP LGM simulations, Siberia receives less precipitation; however, we do not find indications that the buildup of a Siberian ice sheet was hampered by the absence of precipitation." From modelling side, we already know that the precipitation is not the key forcing for understanding glaciations over Beringia.

3) Why is a good simulation of LGM Laurentide-Eurasia-only ice sheet a pre-condition for understanding climate over Beringia? One example is the simulation from Abe-Ouchi et al., 2013. This simulation well reproduces the Laurentide-Eurasia-only ice sheet in LGM. However, this simulation shows that NE Siberia-Beringia is totally ice-free during the past four glacial-interglacial cycles, without any mountain glaciers or ice on the continental shelf, in conflict with direct evidence. This simulation is a good example that does not support your suggestion.

We have to admit that the interpretations of direct glacial or climate evidence from NE Siberia-Beringia are highly controversial. Please allow us to point out the uncertainties in the perennial lake-ice explanation in Lake El'gygytgyn. Lacking absolute age controls and based on a tuned age model, it is difficult or uncertain to identify gaps within these sediments in glacial facies. When we review the high-resolution magnetic susceptibility stratigraphy from core PG1351, we at least can find gaps at ∼310 cm (tuned in MIS4), ∼680 cm (tuned in MIS6c-b), ∼900 cm (tuned in MIS6e). In Site 5011-1, the sediments in glacial facies include turbidites (Sauerbrey et al., 2013). In a permanent frozen land and lake, what process causes these gaps and mass movements? It is obvious that these gaps and mass movements bring uncertainties in the lake-ice explanation, but often neglected. The second example is the evidence you listed in the review, the mammoths' fossils on Wrangel Island from 48 ka to nearly 3400 years ago (Vartanyan et al., 1993). The explanation of this evidence depends on the temporal resolution in age controls. Wrangel Island is in the marginal of the BerIS. On Wrangel Island, ice cover lasts only 1-2 kys. When the Beringian ice sheet reaches its peak, it will melt rapidly, and Wrangel Island exposes. Without age controls in a high temporal resolution, it is difficult to know whether there are small gaps with mammoths' fossils missing on the island. Therefore, it remains uncertain to use ice-free Wrangel Island to indicate the nonexistence of the BerIS. It is possible that Wrangel Island is ice-free, while a substantial BerIS remains on the NE Siberian-Beringian continents. As we wrote in the paper, "partly due to poor age controls, it remains highly controversial whether the glacial evidence points towards a pre-LGM ice sheet over NE Siberia-

Beringia or local activities of ice domes/sheets on continental shelves and mountain glaciers on continents."

Did a BerIS once exist? To answer the question, we should put each piece of evidence (not only the direct glacial evidence) into one framework, without conflicts. Since the direct evidence cannot really answer the question, we turn our eyes to investigate the continuous climate records with precise age controls.

Our simulations indicate that the Laurentide-Eurasian ice sheets alone (even the ICE6G reconstructed) fail in explaining these paleoclimate records from around the North Pacific, whereas the fast waxing and waning of the BerIS success. This is the added value of our study. The physical mechanism behind is the interactions between ice sheets and mid-latitude climates, which are revealed in Fig 4.

It is easy, but quite unfair, to use modelling uncertainties to reject a modelling study. When compared to the map (Figure 63.2) from Glushkova, 2011, and the map (Figure 33.1) from Kaufman et al., 2011, our simulated BerIS is reasonable. However, simulations always include unavoidable uncertainties. These uncertainties should be improved and further constrained with more models in future. "Although the above uncertainties in the ice sheet modelling should be revisited in future studies to archive more realistic simulations for past ice sheet evolutions, these uncertainties do not influence the main logic in this study." Our purpose is not to unequivocally resolve the ice sheet limits, but to find which ice sheet scenario can well reconcile climate evidence from around the North Pacific.

In summary, there are many conflicts in the current interpretations of direct glacial evidence over NE Siberia-Beringia and the mainstream concept of Laurentide-Eurasia-only ice sheets. "The reconciliation cannot be achieved through the growth of ice domes on the NE Siberian continental shelf or mountain glaciers on the NE Siberian continent, since the small-scale glaciations across NE Siberia-Beringia cannot cause strong climate feedbacks to match the paleoclimate records from around the North

Pacific.

It is now too early and simplified to reject the possibility of the BerIS, before these conflicts are well resolved in the concept of Laurentide-Eurasia-only ice sheets together with mountain glaciers over NE Siberia-Beringia.

If you have more criticisms or comments, please let us know.

Regards

Zhongshi on behalf of all co-authors

1. Abe-Ouchi, A., Saito, F., Kawamura, K., Raymo, M.E., Okuno, J., Takahashi, K., Blatter, H.: Insolation-driven 100,000-year glacial cycles and hysteresis of ice-sheet volume. Nature 500, 190-193, 2013. 2. Bakker, P., Rogozhina, I., Merkel, U., Prange, M.: Hypersensitivity of glacial temperatures in Siberia. Climate of the Past, 16, 371-386, 2020. 3. Glushkova, O. Y. Late Pleistocene glaciations in north-east Asia//Developments in Quaternary Sciences. Elsevier 15, 865-875, 2011. 4. Kaufman, D.S., Young, N.E., Briner, J.P., Manley, W.F.: "Alaska Palaeo-Glacier Atlas (Version 2)" in Quaternary Glaciations Extent and Chronology, Part IV: A Closer Look, Developments in Quaternary Science, J. Ehlers, P.L. Gibbard, P.D. Hughes, Eds, (Elsevier, 2011), pp.427-445. 5. Melles, M., Brigham-Grette, J., Minyuk, P. S., Nowaczyk, N. R.,Wennrich, V., DeConto, R. M., Anderson, P. M., Andreev, A.A., Coletti, A., Cook, T. L., Haltia-Hovi, E., Kukkonen, M.,Lozhkin, A. V., Rosén, P., Tarasov, P. E., Vogel, H., and Wag-ner, B.: 2.8 Million Years of Arctic Climate Change from Lake El'gygytgyn, NE Russia. Science 337, 315-320, 2012. 6. Nowaczyk NR, Melles M, Minyuk P (2007) A revised age model for core PG1351 from Lake El'gygytgyn, Chukotka, based on magnetic susceptibility variations tuned to northern hemisphere insolation variations. J. Paleolimnol., 37, 65–76, 2007. 7. Sauerbrey, M. A., Juschus, O., Gebhardt, A. C., Wennrich, V., Nowaczyk, N. R., and Melles, M.: Mass movement deposits in the 3.6 Ma sediment record of Lake El'gygytgyn, Far East Russian Arctic, Clim. Past, 9, 1949–1967, https://doi.org/10.5194/cp-9-1949-2013, 2013.

---

## Short Comment (SC1) · 24 May 2020

Zhang et al. (line 278) state, "The simulated extent [of the Beringian ice sheet] agrees nicely with the mapped distribution of glacial landforms..." including in Alaska for MIS4 and 6. I do not expect a numerical ice-sheet model to correctly simulate the details of the observational evidence of former glacier extent, and it is comforting that the simulation leaves some of the SW part of the state free of ice. However, asserting that the data and the model "agree" is a serious mischaracterization. A random sampling of sites in the two reconstructions to test whether they were covered by ice no not would likely fail statistically to show what is clear visually (Fig. 1, below) – the data

[Figure]

and the models do not agree. Contrary to the authors' conclusion (line 469) that, "In summary, whether a pre-LGM BerIS once existed remains to be an open question," and despite their pointing to the glacial geological evidence in support of their model, the observational and geochronological data preclude an ice sheet in Alaska outside of the Cordilleran during MIS4 and 6.

Kaufman, D.S., Young, N.E., Briner, J.P., Manley, W.F., 2011. Alaska Palaeo-Glacier Atlas (Version 2). In: Ehlers, J., Gibbard, P.L., and Hughes, P.D. (eds), Quaternary Glaciations Extent and Chronology, Part IV: A Closer Look. Developments in Quaternary Science 15, Amsterdam, Elsevier, 427–445.

Zhang et al. (in review) Simulated Beringian ice sheet

[Figure]

Kaufman et al. (2011) Alaska Paleo-Glacier Atlas

[Figure]

FIGURE 33.1   Alaska Palaeo-Glacier Atlas v2 showing the Pleistocene maximum, early Wisconsinan, late Wisconsinan and modern glacier extents across Alaska.

**Fig. 1.**

---

## Author Comment (AC3) · 26 May 2020

Dear Darrell,

Thanks for reading and commenting on our paper.

We agree with you that our simulated ice extent in Alaska still includes mismatches with regional direct evidence, due to the modelling uncertainties and the coarse resolution of the climate model. However, our simulations reveal the following two points, which agree reasonably with your map. 1) Alaska is not fully covered by ice during past glacial-interglacial cycles. 2) The ice or glacier system on Alaska can be divided into

two parts, one in the north, the other in the south. The south part is connected to Cordilleran ice sheet, also shown in your map. Our simulations suggest that the north part belongs to the BerIS. Alaska is the key region to investigate whether the BerIS and the Cordilleran ice sheet was once connected or not.

On the other hand, when we consider the map from Glushkova (2011) and your map in Alaska (Kaufman et al, 2011) together, it remain acceptable to say that our simulated BerIS agrees nicely, at least reasonably, with these maps. The majority of the BerIS is located on the NE Siberian side, not in Alaska. The ice sheet on the NE Siberian continents plays a crucial role in reorganizing the atmosphere circulations.

Very detailed reconstructions for the regional ice sheet extent in Alaska relies on the coupled high-resolution climate and ice sheet models, and often needs down-scaling simulations. When the possibility of the BerIS can be reconsidered, we believe there will be more efforts to constrain the ice extent in Alaska with high-resolution models.

If you have more comments, please let us know.

Zhongshi on behalf of all coauthors

1. Glushkova, O. Y. Late Pleistocene glaciations in north-east Asia//Developments in Quaternary Sciences. Elsevier 15, 865-875, 2011.

2. Kaufman, D.S., Young, N.E., Briner, J.P., Manley, W.F.: "Alaska Palaeo-Glacier Atlas (Version 2)" in Quaternary Glaciations Extent and Chronology, Part IV: A Closer Look, Developments in Quaternary Science, J. Ehlers, P.L. Gibbard, P.D. Hughes, Eds, (Elsevier, 2011), pp.427-445.

---

## Short Comment (SC2) · 27 May 2020

Dear Zhongshi Zhang and co-authors,

Originally, we did not want to step into the discussion of your manuscript, since the conclusions drawn are so obviously in conflict with the geological evidence obtained over the past decades from land records that we thought it is not necessary. However, your replies to the two constructive reviews the manuscript has received push us to do so, since they illustrate that you do not understand or do not want to understand and consider major geological findings.

[Figure]

In our comment we would like to stick to the record of Lake El′gygytgyn, which according to your simulations has repeatedly been covered by an ice sheet that reached a thickness of more then 2 km over the lake at least during MIS 4 and MIS 6. During these times, Lake El′gygytgyn received laminated sediments, which contain organic matter and fossils, are not overconsolidated and do not contain glacial debris, and thus are interpreted as lacustrine sediments deposited during glacial times (therefore named ′glacial facies′) in a stratified water body underneath perennial lake ice (Melles et al. 2007, 2012, Wennrich et al. 2016, and various other papers). Let us stress just a few of your statements concerning these intervals.

(1) In the reply to the review of Julie Brigham-Grette you question the age models of cores PG1351 and ICDP 5011-1 from Lake El′gygytgyn, making it "difficult or uncertain to identify gaps within these sediments". Fact is that the age models for these cores are not just "tuned age models", they are also based on luminescence and paleomagnetic dating (e.g., Nowaczyk et al. 2007, 2013, Melles et al. 2012), complemented and confirmed by respective data and 14C ages from additional cores (Lz1024, Lz1029, Lz1039, Lz1041; e.g., Juschus et al. 2007, 2009). From these results it is clear that deposition in the lake continued during glacial stages via well known processes and that the glacial sediments kept preserved.

(2) In the same reply letter you then identify several gaps based on the magnetic susceptibility data of core PG1351 (without explaining this assumption or giving a reference for that) and question: "In a permanent frozen land and lake, what processes causes these gaps and mass movements?". First, neither the land nor the lake were completely frozen (e.g., Melles et al. 2007, Schwamborn et al. 2012 and many many more papers on the subaquatic deposition and permafrost active-layer behavior through time). Even with a perennial ice-cover, sediment influx occurred during the summer season via the formation of moats along the shores (as known, for instance, from the McMurdo Dry Valleys of the Antarctica; Warwick et al. 2008). Second, the minor gaps in the MS record of core PG1351 are due to the deletion of loop sensor

data at the ends of core segments, not to gaps in the sediments, as could be shown by the correlation of PG1351 with parallel core Lz1024 and deep-drilling core ICDP 5011-1, resulting in a 100% recovery of the Quaternary sequence (see Melles et al. 2011, Frank et al. 2013). And third, mass movements indeed occur throughout the record, although mainly during interglacial times, but in the central part of Lake El′gygytgyn they were associated with minor if any erosion, at least in the time frame of the glaciations simulated by you (Juschus et al. 2009, Sauerbrey et al. 2013, Warnke et al. 2020).

(3) In the reply to reviewer 1 you do not argue against continuous sedimentation during the deposition of the ′glacial facies′ in Lake El′gygytgyn. Instead, in this reply you speculate that "Lake El′gygytgyn could be a subglacial lake when there is an ice sheet on it". This is in clear conflict for instance with the concentrations of organic carbon and the sedimentation of pollen during these periods, and in particular with the accumulation of diatoms, which cannot be supplied by the ice sheet, taking their fragility and the lack of exclusively freshwater sediment sources for the ice sheet. A formation under a floating ice sheet also is not possible, taking the lack of light requested by photosynthesis-performing diatoms.

(4) Finally, we wonder why you do not mention with any word (in the manuscript as well as in both replies to the reviews) the climate modeling that was already carried out to obtain a better understanding of the sediment proxies measured on the Lake El′gygytgyn record (e.g., Melles et al. 2012). In this work numerical climate simulations involving changes in orbital configurations, greenhouse gas concentrations, and continental ice coverage were also not always able to match the climate history evidenced by pollen data in Lake El′gygytgyn, probably due to changes in the oceanic circulation that originate in Antarctica. Why are you so confident that the simulations you run really consider all natural processes and feedback mechanisms that may have occurred in the wide time frame you are dealing with?

We hope with these comments we have made clear enough that you are wrong with your statement that "the glacial sedimentological facies in Lake El′gygytgyn (Melles et

al. 2012) ... needs reinterpretation". As things go: Lake El′gygytgyn by chance is located in the centers of the Beringian ice sheets simulated by your group. This is unfortunate for your model results, but pleasant for science and hopefully will convince the community again that glacial ice coverage in that region was much more limited than suggested by you, as mapped by generations of geologists and geomorphologists.

Regards

Martin Melles and Volker Wennrich (University of Cologne)

\_\_\_\_\_\_\_\_\_\_\_\_

References:

Juschus O., Preusser F., Melles M. & Radtke U. (2007): Applying SAR-IRSL methodology for dating fine-grained sediments from Lake El′gygytgyn, northeastern Siberia. - Quaternary Geochronology, 2: 137-142.

Juschus O., Melles M., Gebhardt A.C. & Niessen F. (2009): Late Quaternary mass movement events in Lake El′gygytgyn, northeastern Siberia. - Sedimentology, 56: 2155-2174.

Forman S.L., Pierson J., Gomez J., Brigham-Grette J., Nowaczyk N.R. & Melles M. (2007): Luminescence geochronology for sediments from Lake El′gygytgyn, northwest Siberia, Russia: Constraining the timing of paleoenvironmental events for the past 200 ka. - Journal of Paleolimnology, 37: 77-88.

Frank U, Nowaczyk N.R., Minyuk P., Vogel H., Rosén P. & Melles M. (2013): A 350 ka record of climate change from Lake El'gygytgyn, Far East Russian Arctic: refining the pattern of climate modes by means of cluster analysis. - Climate of the Past, 9: 1559-1569.

Melles M., Brigham-Grette J., Glushkova O.Yu., Minyuk P., Nowaczyk N.R. & Hubberten H.-W. (2007): Sedimentary geochemistry of a pilot core from El′gygytgyn Lake

[Figure]

- a sensitive record of climate variability in the East Siberian Arctic during the past three climate cycles. - Journal of Paleolimnology, 37: 89-104

Melles M., Brigham-Grette J., Minyuk P., Koeberl C., Andreev A., Cook T., Fedorov G., Gebhardt C., Haltia-Hovi E., Kukkonen M., Nowaczyk N., Schwamborn G., Wennrich V. & the El'gygytgyn Scientific (2011), The Lake El'gygytgyn Scientific Drilling Project – Conquering Arctic Challenges through Continental Drilling. - Scientific Drilling, 11 29-40.

Melles M., Brigham-Grette J., Minyuk P.S., Nowaczyk N.R., Wennrich V., DeConto R.M. Anderson P.M., Andreev A.A., Coletti A., Cook T.L., Haltia-Hovi E., Kukkonen M., Lozhkin A.V., Rosén P., Tarasov P., Vogel H. & Wagner B. (2012): 2.8 million years of Arctic climate change from Lake El'gygytgyn, NE Russia. - Science, 337: 315-320.

Nowaczyk N.R., Melles M. & Minyuk (2007): A revised age model for core PG1351 from Lake El′gygytgyn, Chukotka, based on magnetic susceptibility variations correlated to northern hemisphere insolation variations. - Journal of Paleolimnology, 37: 65-76.

Nowaczyk N.R., Haltia E.M:, Ulbricht D., Wennrich V., Sauerbrey M.A., Rosén P., Vogel H., Francke A., Meyer-Jacob C., Andreev A.A. & Lozhkin A.V. (2013): Chronology of Lake El'gygytgyn sediments – a combined magnetostratigraphic, palaeoclimatic and orbital tuning study based on multi-parameter analyses. - Climate of the Past, 9, 2413–2432.

Sauerbrey M.A., Juschus O., Gebhardt A.C., Wennrich V., Nowaczyk N.R. & Melles M. (2013): Mass movement deposits in the 3.6 Ma sediment record of Lake El'gygytgyn, Far Easst Russian Arctic. - Climate of the Past, 9: 1949-1967.

Schwamborn G., Schirrmeister L., Frütsch F. & Diekmann B. (2012): Quartz weathering in freeze–thaw cycles: experiment and application to the El'gygytgyn crater lake record for tracing siberian permafrost history. - Geografiska Annaler: Series A, Physical Geography, 94:4, 481-499.

Warnke F., Gebhardt C. & Niessen F. (2020): Glacial-interglacial cycles largely controlled mass movements during the late Quaternary in Lake El'gygytgyn, Siberia. - Palaeogeography, Palaeoclimatology, Palaeoecology, 539 109506.

Wennrich V., Andreev A.A., Tarasov P.E., Fedorov G., Zhao W.W., Gebhardt C.A., Meyer-Jacob C., Snyder J.A., Nowaczyk N.R., Chapligin B., Anderson P.M., Lozhkin A.V., Minyuk P.S., Koeberl C. & Melles M. (2016): Impact processes, permafrost dynamics, and climate and environmental variability in the terrestrial Arctic as inferred from the unique 3.6 Myr record of Lake Elgygytgyn, Far East Russia - a review. - Quaternary Science Reviews, 147: 221-244.

Vincent W.F., MacIntyre S., Spigel R.H. & Laurion I. (2008): The physical limnology of high-latitude lakes. - In: W.F. Vincent & J. Laybourn-Parry (Eds.), Polar Lakes and Rivers. Oxford University Press, New York, NY, p. 65-81.

---

## Author Comment (AC4) · 28 May 2020

Dear Martin and Volker,

We agree that the age model from Lake El'gygytgyn is the best solution available now. However, unavoidable uncertainties are included in the age model. Thus, the age model was adjusted in 2002, 2007, 2013 (Nowaczyk et al., 2013). Moreover, the gaps and mass movements in glacial facies from Lake El'gygytgyn should be further investigated, since they bring uncertainties in explaining glacial sediments from Lake El'gygytgyn.

[Figure]

Due to these uncertainties, the interpretation of the glacial facies remains controversial. Based on the concept of Laurentide-Eurasia-only ice sheets (without the BerIS), the perennial lack-ice explanation is reasonable. On the contrary, based on a new scenario with the BerIS involved, the subglaical lake explanation is not unacceptable. The subglacial lake can stop receiving sediments (or sediments are eroded or reworked by ice) when the BerIS is very big, while receive sediments again when the ice melts.

As we wrote in the reply to Juile, "Did a BerIS once exist? To answer the question, we should put each piece of evidence (not only the direct glacial evidence) into one framework, without conflicts. Since the direct evidence cannot really answer the question, we turn our eyes to investigate the continuous climate records with precise age controls."

In our study, with the same climate model, we test the two ice sheet scenarios to investigate which one could reconcile the climate (temperature) evidence from around the North Pacific. Unfortunately, the scenario of Laurentide-Eurasia-only ice sheets fails. To make our simulations convincing, we validate our climate model by carrying out sensitivity experiments forced with the ICE6G reconstructions. Forced by the Laurentide-Eurasia-only ice sheets, our sensitivity experiments demonstrate that the simulated large-scale atmosphere and ocean responses agree with early modelling studies. Please see lines 236-256 in the paper.

To further strengthen the concept of Laurentide-Eurasia-only ice sheets, there are two questions that should be answered.

1) What forcing limits the growth of the BerIS? As proved by our and the early study (Bakker et al., 2020), the buildup of a BerIS is not hampered by the absence of precipitation.

2) How to reconcile the temperature evidence from around the North Pacific within the Laurentide-Eurasia-only concept?

Before these two questions are well answered, it is not fair to reject the possibility of the BerIS now.

We suggest that whether a pre-LGM BerIS once existed remains to be an open question. We hope our current work can encourage more studies in the Beringian regions. From modelling side, we need to "further distinguish the climate feedbacks due to the BerIS and the Laurentide-Eurasia-only ice sheets" with more climate models. We suggest that "experiments of MIS 4 could be a new benchmark in the PMIP". From data side, "new field and marine field investigations across NE Siberia-Beringia, to acquire sea level sequences, glaciostatic changes, and paleoclimate records in the Beringian regions, are clearly key targets to provide more precise age controls and robust constraints to the extent and timing of the BerIS."

If you have more comments, please let us know.

Zhongshi on behalf of all co-authors

———————————————

---

## Referee Comment (RC3) · Lev Tarasov (Referee) · 7 Jun 2020

The submission by Zhang et al, sets out to challenge the stated
paradigm of no large scale glaciation of Eastern Siberia during the
last few (?) glacial cycles. To progress, science needs such
challenges. However, in this case the challenge is carried out with an
inadequate model setup that relies on surface melt parameters that are
hard to defend (or not defensible at all for the IDL
configuration). These parameter values reduce surface melt, and yet
the model is still unable to get adequate LGM ice volume in the
Northern Hemisphere.  Furthermore, the presentation is often imprecise,
and at times makes claims/comparison in reference to the litterature
that are not supported by the given citation. The model description is
also inadequate, and relies on boundary conditions that are not
available from the cited source (ICE6G prior to 26 ka).

I would therefore recommend rejection and encourage the authors to:

1) revise the PISM setup to reasonable PDD melt factors (anything below
the commonly used 3.0 and 8.0 mm/PDD water equivalent would need clear
justification) and appropriate model resolution (if PISM doesn't have
a subgrid massbalance/flow module, then run no coarser than 10 km,
preferably at 5 km resolution if you want to have any confidence in
Alaskan results). Extract the standard deviation of hourly
temperatures (relative to the presumably monthly mean output passed to
PISM) from CAM.

2) Ideally run CAM at T85. If not feasible, then explicitly
consider resolution related uncertainties from the litterature
to impose a resolution correction to your climate fields

3) provide adequate model description

4) make more careful and precise comparisons to the litterature and
do not mis-represent the litterature

5) show the present-day ice sheet configuration with your chosen
setup (to show you have not biased your model to a cold and/or reduced
melt configuration)

**detailed comments**

**Yes, but you should mention that limited local glaciation (eg alpine**
**and valley glaciers) is inferred ( eg Elias and Brigham-Grette,**
**2013 (DOI: 10.1016/B978-0-444-53643-3.00116-3), who show multiple maps**
**from cited litterature of Eastern Siberian glaciation.**
One concept widely held is that during most glacials only the
Laurentide-Eurasian 32 ice sheets across North America and Northwest
Eurasia became expansive, while Northeast Siberia-Beringia 33 remained
ice-sheet-free

**should also cite Andrews (1992, Nature) as well as Clark and Tarasov**
**(2014, PNAS) concerning the LGM missing ice issue**
A comparison between estimations of 68 Laurentide-Eurasian ice sheet
volume and direct observations of sea level change during the LGM
reveals a 69 discrepancy of unexplained missing ice with a volume of
~6-25 m ice-equivalent sea-level change (Simms 70 et al., 2019).

**This is not at all unexpected. Just because temperatures start to**
**rise, doesn't mean ice volume can't keep increasing for a certain**
**amount of time.**
The 122 DH δ18O records show that towards the end of each of the last
four full glacial cycles, the mean surface 123 temperature started
increasing earlier in terrestrial regions on the mid-latitude North
American west coast, 124 while the NH ice volume kept increasing
(Fig. 1c). Such

**ICE6G has no pre-LGM constraint, so I don't see how this is a**
**relevant boundary condition**
In the result section below, our simulations forced with the ICE6G ice
sheet reconstructions (Peltier et 138 al., 2015)will investigate
whether the growth of the Laurentide-Eurasian ice sheets alone can
explain the 139 early warming and the asymmetry changes from around
the North Pacific.

**It is already well documented in the modelling litterature that T31**
**(and to a still significant extent T42) does not adequately resolve**
**atmospheric circulation, especially for the context of simulating**
**Eurasian ice sheets, cf Lofverstrom and Liakka (2018, TC)**
The resolution of spectral CAM4 is approximately 3.75° (T31) in the
147 horizontal and 26 levels in the vertical

**imprecise meaningless statement**
has good skill in simulating paleoclimates (Zhang et al., 2013; 2014).

**what is the chosen relationship?**
The snowfall is determined based on the 165 partitioned total
precipitation following an empirical relationship relating total
precipitation and air 166 temperature.

**This resolution is too coarse for the accurate modelling in**
**topographically complex regions such as Alaska, which look like a**
**bumpy plateau at this resolution. With currently available standard**
**computational resources, there is also no justification for relying**
**on an ice sheet model at only such a coarse resolution unless you**
**are doing large ensembles of a Myr or longer. I would want to see no**
**coarser than 20 km resolution along with a subgrid massbalance/flow**
**model such as that of Marshall, 2002 (QI), or LeMorzadec et al, 2015**
**(GMD) to have any confidence in modelling of Alaska.**
three-dimensional, thermodynamically coupled continental-scale hybrid
ice sheet 160 model (Winkelmann et al., 2011; Martin et al., 2011; The
PISM authors, 2015), run at a resolution of 40 161 km×40 km in this
study

**These are very low PDD melt factors and standard deviation compared**
**to other modelling studies as well as observations. Hock (2003**
**J. Hydrol.)  in her Table 1, lists PDD melt factors. The lowest**
**values for glacial conditions are 2.7 and 5.4 mm/PDD (melt water**
**equivalent) with averages above 4.0 and 8.0 mm/PDD.  The unjustified**
**lack of SIA enhancement will also promote thick ice favouring your**
**hypothesis, though this depends on what ice rheology relationship**
**you are using and which you do not provide. As stated (daily cycle),**
**are you implying that you use daily mean temperatures from the GCM**
**in PISM?  Or are you using a more standard mean monthly or mean**
**monthly climatology?  It would also make more sense to extract the**
**standard deviation from CAM than chose a relatively low value of 2.5**

**K.**
**This low melt biasing of your setup is evident is clearly evident in**
**the excessive MIS3/2 Beringian ice shown in your supplemental fig 5**
Here, we set the daily melt rate to 5 mm/doC for ice 168 (PDD_ice),
and 2 mm/doC for snow (PDD_snow), with a standard deviation of 2.5 °C
for the daily cycle of 169 surface air temperature (Temp_std). Ice
velocities are modulated by means of enhancement factors set to 1 170
for flow treated with SIA (ENF_SIA), and 0.1 for flow treated with SSA
(ENF_SSA)

**write down the power law. A reader should not have to refer to**
**another paper to understand core components of the experimental**
**setup.**
Basal sliding is based on 173 a pseudo-plastic power law model (Greve
and Blatter, 2009) in which the exponent q is set to 0.25 174
(pseudo_plastic_q).

**downscaling of GCM fields to PISM grid needs to be described.**

**close in what sense? Your modelling LGM ice volume is evidently way**
**too small.**
 Laurentide
181 ice sheet close to reconstructions

**These parameters are totally unjustified.  Observed PDD melt factors**
**are never this low. And do you really expect eg that the standard**
**deviation of hourly temperatures around the monthly (or even daily**
**mean) is 1 K?**
We set PDD_ice to 2 mm/d°C, PDD_snow to 1 mm/d°C, Temp_std to 1 °C,
183 ENF_SIA to 1, ENF_SSA to 0.1

**what present-day extent and ice volume do your chosen configurations**
**give?**

**Peltier's website (as cited in this submission) does not provide**
**ICE6G prior to 26ka, so where does ICE6G-70ka come from?  Nor does**
**the cited Peltier et al, 2015 provide any results prior to 26 ka.**
The comparison
201 between the ICE6G-22ka (ICE6G-70ka)

SH ice sheet extent is fixed and uses the modern condition
**why? Why not use ICE6G for SH as well?**

**at some point need to discuss the issue that ice sheets were**
**unlikely to be in equilibrium with climate**
use this simulated climate to force PISM to get the NH ice sheets in
equilibrium 211 with the simulated climate of 126 ka

**It clearly does not for Alaska if I compare figure 5 to fig 33.1**
**all-time glacial extents in Kaufman et al., 2011 as Darrell Kaufman**
**has already made clear**
The simulated extent (Fig. 5, 7 and Supplementary Fig.5, 6) agrees 279
nicely with the mapped distribution of glacial landforms across NE
Siberia-Beringia (Stauch and Gualtieri, 280 2008; Darrell et al.,
2011; Kaufman et al., 2011; Glushkova, 2011; Barr and Clark, 2012a,b;

Niessen et al., 281 2013; Barr and Solomina, 2014; O'Regan et al., 2017; Nikolskiy et al., 2017; Tulenko et al., 2018; Batchelor

**Are you now claiming that there was this much ice in this region at**
**LGM? If not, then what is the point of this statement?**
282 et al., 2019). The simulated ice volume accounts for ~10-25 m ice-equivalent sea-level change (~20-30% or 283 more of simulated NH ice volume, Supplementary Fig. 5), coinciding with the volume of the missing ice 284 during the last glacial (Simms et al., 2019)

**This validation is very limited and "realistically simulates" is a judgment you are making**
**that ignores the issue I raised above concerning required minimum resolution to adequately**
**resolved atmospheric circulation changes.**
Second, we validate the climate model, 319 NorESM-L, and show it realistically simulates the climate responses caused by the Laurentide-Eurasian ice 320 sheets (Fig. 4a-d), in agreement with earlier studies

**not for Alaska**
The simulated BerIS (Fig.  325 7) agrees reasonably with the direct glacial evidence across NE Siberia-Beringia

**Even with a subglacial lake present, the normal stress from surface**
**loading should be evident in the sediment structure (extent of**
**consolidation).  Furthermore, a subglacial lake would indicate warm**
**based conditions. And this lake could not have been continuously**
**maintained right through deglaciation. At some point, drainage of the**
**lake would have placed warm based ice on top of the warm near surface**
**sediments, resulting in some subglacial till deformation if there were**
**any significant driving stress on the ice**
cp-2020-38-AC1-supplement.pdf : Lake El'gygytgyn could be a subglacial lake when there is an ice sheet on it.

---

## Author Comment (AC5) · 8 Jun 2020

Dear Lev,

Thanks for reading and commenting on our paper.

We agree with you to precisely simulate an ice sheet needs to find good parameters for an ice sheet model and to use high-resolution climate models. This is our future task. However, the motivation of our current paper is not to unequivocally resolve the ice sheet limits.

Our motivation is to test which ice sheet scenarios can well explain the climate evidence

from around the North Pacific, the Laurentide-Eurasia-only ice sheet scenario, or the scenario with the BerIS involved. In our simulations, we demonstrate that, without the BerIS or only with mountain glacials over NE Siberia-Beringia, the Laurentide-Eurasia-only ice sheet scenario (even the widely used ICE6G reconstructions) fails.

Without answering the question why the Laurentide-Eurasia-only ice sheet scenario can reconcile the climate evidence from around the North Pacific, it is quite unfair to reject the possibility of the BerIS only based on uncertainties in ice sheet modelling. We think a fair rejection should point out the mistakes in the reconciliation within the BerIS scenario and the climate evidence from around the North Pacific revealed in our current study.

We fully understand that the Laurentide-Eurasia-only ice sheet scenario is the mainstream concept today. However, to further strengthen this concept, there are two questions that should be answered. 1) What forcing limits the growth of ice sheet over NE-Siberia, since the buildup of an ice-sheet there is not hampered by the absence of precipitation. 2) How to reconcile the temperature evidence from around the North Pacific within the Laurentide-Eurasia-only concept?

We appreciate that you agree science needs such challenges. However, to reject the possibility of the BerIS now will make few scientists be willing to rethink above questions.

Best regards

Zhongshi Zhang

---

## Referee Comment (RC4) · Julie Brigham-Grette (Referee) · 10 Jun 2020

I believe you continue to miss the point. You are proposing to place an ice sheet over places where other styles of sedimentation were taking place throughout glacial/interglacial cycles, and where a large body of other evidence reveals that the region surrounding Lake El'gygytgyn could not have experienced continental glaciation (Melles et al, 2012; Brigham-Grette et al, 2013). Your response seems to prefer criticizing my suggestions instead of replying to the real problem. The real problem is placing an ice sheet over areas known to be ice free without continental scale ice sheets (Bond, 2019 compilation). Your view of modeling an ice sheet that agrees with

stalagmite in Nevada does not mean you can simply discard all of the cosmogenic isotope ages, radiocarbon dates, and other geochronology for lacustrine and glacial sequences in Beringia (Goetcheus and Birks, 2001; Hamilton, 2001; Brigham-Grette et al. 2003, for example).

The Lake El'gygytgyn record is a continuous sedimentological sequence recording the last 3.6 million years. The sedimentology does have primary facies that we described (Melles et al 2012; Brigham-Grette et al, 2013; Wei et al, 2014), and our conclusions of past climates and depositional environments inferred from these facies are supported by a suite of other independent proxies including total organic carbon, biogenic silica (productivity), diatom taxonomies, and pollen reflecting both cold and warm intervals. Biomarkers, (esp. branched glycerol dialkyl glycerol tetraethers or brGDGTs) produced in the lake by bacteria, have been analyzed at a resolution of ∼2,000 years over the length of the entire 3.58 Ma (Holland et al., 2013; D'Anjou et al. 2013; De Wet et al. 2016, Keisling et al., 2017; Castañeda et al, in prep; Daniels et prep) and the leaf wax deuterium isotope record of precipitation is also continuous at a varying resolution of 1 to 5 ka years through glacial and interglacial periods back to 280ka (Habicht et al., in prep). Furthermore, there is new research emerging that shows Lake El'gygytgyn contains evidence of fire in NE Russia (black carbon, PAHs and levoglucosan). And fire appears to be common during cold dry glacial intervals (McConnell et al, in prep). Fires can't ignite under an ice sheet.

A large body of evidence representing research conducted during the past 20 years has all arrived at the same conclusion; that Lake El'gygytgyn has continuously accumulated sediment, including algal and terrestrial organic constituents, since its formation (See, for example papers in Special issue in Climate of the Past on Lake El'gygytgyn). One ice sheet/climate modeling study should not simply ignore such a large body of physical, chemical and biological evidence simply because it does not agree with the output.

The total of the sedimentology and the other proxies confirm our interpretation that this

record is without gaps, precluding the repeated expansion of a 2 km thick ice sheet over the area. We have several peer-reviewed papers describing how the turbidite events were identified and removed from the composite record (see Warnke et al 2020 and references therein). The lake record is documented by over 70 peer reviewed papers focused on the evidence from the lake sediments alone. The chronology of the initial pilot cores that are continuous to about 250ka were dated by infrared stimulated luminescence and radiocarbon, (Forman et al, 2007; Juschus et al., 2007, 2009). There are 8 volcanic ashes preserved in the lake sequence, with the youngest dated to the late Pleistocene (Juschus et al. 2009; van den Bogaard, et. al. 2014).

Aridity cross Beringia was widespread especially in MIS 2 and there are papers suggesting the presence of richer vegetation in riparian areas within a mosaic of landscapes (see Miller et al, 2010 and references there in). You seem to dismiss the glacial record of well-dated moraines in Chukotka, the Brooks Range and Seward Peninsula where MIS 6 (and sometimes MIS 8 and 10) are typically more extensive than MIS2; see the maps (http://instaar.colorado.edu/QGISL/ak_paleoglacier_atlas/gallery/index.html). As suggested in my first review, I strongly suggest you allow your model to correctly reproduce the MIS 2 ice extent (see earlier review for map of Jeff Bond), and the moraines we can visit for MIS 6, then evaluate what an ice sheet in the East Siberian Sea would look like. I maintain that good science considers all of the evidence. You seem to be letting your model control what evidence makes sense. You should consider adapting a multi-model approach that allows for a more critical view of model-dependent output vs. field evidence (see for example Alder and Hostetler, 2017)

Alder, J.R., and Hostetler, S.W., 2017. Application of an ice sheet model to evaluate PMIP3 LGM clinatologies over the North Amercian ice sheets. Climate of the Past Clim. Past Discuss., https://doi.org/10.5194/cp-2017-102

Brigham-Grette, Julie, Gualtieri, L.M., Glushkova, O.Yu., Hamilton, T.D., Mostoller, David, and Kotov, Anatoly, 2003, Chlorine-36 and 14C chronology support a limited

last glacial maximum across central Chukotka, northeastern Siberia, and no Beringian ice sheet: Quaternary Research, v. 59, p. 386–398

Goetcheus, V.G., and Birks, H.H. (2001). Full-glacial upland tundra vegetation preserved under tephra in the Beringia National Park, Seward Peninsula, Alaska. Quaternary Science Reviews 20: 135-147.

Hamilton, T.D., 2001, Quaternary glacial, lacustrine, and fluvial interactions in the western Noatak basin, northwest Alaska: Quaternary Science Reviews, v. 20, p. 371–391.

Holland, A. R., Petsch, S. T., Castañeda, I. S., Wilkie, K. M., Burns, S. J., and Brigham-Grette, J.: A biomarker record of Lake El'gygytgyn, Far East Russian Arctic: investigating sources of organic matter and carbon cycling during marine isotope stages 1–3, Clim. Past, 9, 243–260, https://doi.org/10.5194/cp-9-243-2013, 2013.

Juschus, O., Melles, M., Gebhardt, A.C., and Niessen, F., 2009. Late Quaternary mass movement events in Lake El′gygytgyn, north-eastern Siberia. Sedimentol., 56:2155–2174.doi:10.1111/j.1365-3091.2009.01074.x.

Juschus, O., Preusser, F., Melles, M., and Radtke, U., 2007. Applying SAR-IRSL methodology for dating fine-grained sediments from Lake El'gygytgyn, north-eastern Siberia. Quat.Geochron., 2:187–194, doi:10.1016/j.quageo.2006.05.006

Kaufman, D.S., Porter, S.C., and Gillespie, A.R., 2004, Quaternary alpine glaciation in Alaska, the Pacific Northwest, Sierra Nevada, and Hawaii, in Gillespie, A.R., Porter, S.C., and Atwater, B.F., eds., The Quaternary Period in the United States: Amsterdam, Elsevier Developments in Quaternary Science, v. 1, p. 77–103

Miller GH. and 22 others, 2010. Temperature and precipitation history of the Arctic. QSR 29, 1679-1715.

Vanden Bogaard C. et al., 2014, Volcanic ash layers in Lake El'gygytgyn: eight new regionally significant chronostratigraphic markers for western Beringia. Climate of the Past 10, 1041-1062.

Vartanyan, S., Garutt, V. & Sher, A. Holocene dwarf mammoths from Wrangel Island in the Siberian Arctic. Nature 362, 337–340 (1993). https://doi-org.silk.library.umass.edu/10.1038/362337a0

Vartanyan, S.L., Arslanov, K.A., Karhu, J.A., Possnert, G., Sulerzhitsky, L.D., 2008. Collection of radiocarbon dates on the mammoths (Mammuthus primigenius) and other genera of Wrangel Island, northeast Siberia, Russia. Quat. Res. 70, 51–59
* * *

---

## Author Comment (AC6) · 10 Jun 2020

Dear Julie,

We fully understand that you disagree with the concept of Beringian ice sheet. Thus, you criticize that we put an ice sheet on the region where there is no ice sheet.

However, we must admit that interpretations of direct evidence still include uncertainties. In Beringia, it is quite difficult to get "high temporal resolution" absolute age controls. This difficulty makes the interpretations highly controversial. In particular, we show that the ice sheet waxes and wanes rapidly there. The ice sheet only exists for a

few thousand years on Beringia, then disappears. These fast glaciations and deglaciations make it more difficult to distinguish from sediments whether an ice sheet once developed there. It is easier to find sediments during the deglaciation stages (without ice sheet).

Since it is difficult to get a clear answer from the debate of direct evidence, why not jump out to look at evidence from a third side? This is the motivation of our current study.

We test which ice sheet scenarios can well explain the temperature records from around the North Pacific. These temperature records have high-resolution absolute age controls. If the Beringian ice sheet is totally impossible, we should see that the Laurentide-Eurasia-only scenario can well explain these climate records. Unfortunately, this is not the case.

Regards

Zhongshi

―――――――――――――――――――――

---

## Author Comment (AC7) · 10 Jun 2020

Since the Laurentide-Eurasia-only ice sheet is the mainstream concept today, it not surprised to see that many reviewers suggest rejecting our paper. This mainstream concept has been established for two decades. Some reviewers here are the pioneer scientists who established the concept. We show our respects to all these pioneer scientists who involved the early debate of the Beringian ice sheet (BerIS), as we wrote in the acknowledgements.

However, it is a pity to see that the reviewers who suggest rejecting do not point out any mistakes in our logic arguments, but only challenge ice sheet modelling uncertainties

or discuss controversial interpretations of direct evidence.

As we wrote in the paper, we use four steps to address the debate of ice sheet development during past glacial-interglacial cycles.

1) We review the paleoclimate climate records from around the North Pacific.

2) We validate our climate model (NorESM-L), and show it realistically simulates the climate responses caused by the Laurentide-Eurasia-only ice sheets.

3) We use the NorESM-ICE6G experimental flow to indicate that the Laurentide-Eurasian ice sheets alone cannot explain these paleoclimate records from around the North Pacific

4) We use the NorESM-BIOME4-PISM experimental flows to reveal that these climate records and glacial evidence across NE Siberia-Beringia are well reconciled, when the fast waxing and waning of the BerIS are involved.

It is not surprised to see many reviewers do not like the idea of the BerIS. However, if the reviewers believe that the existence of the BerIS is wrong, they should at least let us know why they think the Laurentide-Eurasia-only ice sheets can explain the temperature records from around the North Pacific. We agree with reviewers that ice sheet modelling unavoidably includes uncertainties, which should be further constrained in future, as we have discussed in the paper. However, these uncertainties do not influence the major conclusion in our current study.

In the open discussion, we ask reviewers the following crucial questions many times.

1) What forcing limits the growth of ice sheet over NE Siberia-Beringia, since the buildup of an ice-sheet there is not hampered by the absence of precipitation.

2) How to reconcile the temperature records from around the North Pacific within the Laurentide-Eurasia-only concept?

If the mainstream concept is right, these two questions should be answered easily.

However, we do not see reviewers discuss anything about these two questions.

Before these two questions are well resolved in the Laurentide-Eurasia-only concept, should the possibility of the BerIS be rejected?

I really hope that the BerIS is not a forbidden topic, and further open discussions in future can bring new field-investigations across NE Siberia-Beringia, and new wills for carrying out high-resolution simulations in the region.

---

## Author Comment (AC8) · 18 Jun 2020

In this final response, we focus on two questions. 1) Why do we use the climate records to address the debate of ice sheets during the past glacials? 2) Why do the uncertainties in ice sheet modelling not influence the conclusion of our current study?

Question 1:

As demonstrated in our study, the BerIS waxes and wanes rapidly. The full BerIS only exists for a few thousand years. When the BerIS does not reach its full size or in the deglaciation stage, some regions in NE Siberia can receive sediments that indicate an

environment without an ice sheet.

In other words, without absolute age controls in a "high-temporal-resolution", interpretations of direct evidence on NE Siberia-Beringia still includes uncertainties. We use MIS4 as an example. Suppose we find ice-sheet-free sediments and get one absolute age control within MIS4, we often conclude that the whole MIS4 is ice-sheet-free. However, in NE Siberia-Beringia, a special region, these ice-sheet-free sediments are misleading. These ice-sheet-free sediments can be deposited in the early and late MIS4, but a full ice sheet happens in the middle MIS4. Based on this understanding, we point out the gaps, mass movements or reworked sediments in Lake El'gygytgyn, when we discuss with Juile and Martin.

On the other hand, the scenario with the BerIS waxed and waned rapidly does not have conflicts with these ice-sheet-free evidence. It allows the ice sheet and the ice-sheet-free sediments occurring on NE Siberia-Beringia. Note this possibility was never considered before.

Since the interpretations of direct evidence are highly controversial, it is difficult to resolve the debate only based on direct evidence. Why not look at evidence from a third side? This is the motivation of our current study. We use the temperature records (with precise absolute age controls) from around the North Pacific, and test which ice sheet scenarios can well explain these temperature records. We find that the Laurentide-Eurasia-only scenario fails.

Question 2:

Because the Laurentide-Eurasia-only scenario is the mainstream concept today, when models produce some ice sheets on NE Siberia-Beringia, these simulations are always attributed to model biases or badly selected parameters in ice sheet models, just like the criticisms from Lev. Modelers often face an endless loop, since no simulations are perfect. For example, even an ice sheet simulation is improved, a reviewer who does not like the simulation still can argue that the parameters should be further constrained,

and the climate model resolution needs to be increased. Since this endless loop is not helpful for resolving the debate, why not jump out to look at the evidence from a third side?

In our current study, thanks to the temperature records from around the North Pacific, we use a good climate model to test which ice sheet scenario can reconcile these temperature records, the reconstructed Laurentide-Eurasia-only scenario, or the simulated scenario with the BerIS involved. (Since there are no reconstructions for the BerIS, the scenario with the BerIS must be simulated. We know that we can not unequivocally resolve the ice sheet limits due to the modeling uncertainties). However, with this method (tested with climate records), the ice sheet modelling uncertainties cannot influence the conclusion of our study.

Our climate simulations show that the ICE6G reconstructions cannot explain these temperature records. On the contrary, although the simulated scenario with the BerIS involved still includes uncertainties, this scenario does a better job. It reconciles the temperature records, and do not fall in conflict with direct evidence. In other word, the growth of the Laurentide-Eurasian ice sheet always decreases surface temperature in the mid-latitude North American west coast (both in ocean and on land), while the BerIS is needed to disturb this cooling trend, otherwise the early warming cannot occur there. This is the solid evidence that supports the existence of the BerIS.

We fully agree with Lev that the BerIS should be further constrained with high-resolution climate models and better selected ice sheet parameters. The better constrained BerIS scenario in future can be tested in climate models again by using the method suggested in our current study. They are future tasks that should be considered together by different model groups. However, if the possibility of the BerIS is simply rejected and thought to be wrong, no modelling groups will think it is valuable to consider an ice sheet on NE Siberia-Beringia.

Two years ago, we argued that an ice sheet should exist on NE Siberia-Beringia,

though that pure modelling study was rejected due to ice sheet modelling uncertainties (https://www.clim-past-discuss.net/cp-2018-79/). In that study, we have pointed out the physical mechanism behind. "When only forced with orbital parameters and greenhouse gas levels, changes in atmospheric circulation are weak". Thus, "during some glacials ice sheet expansion starts from a circum-Arctic configuration, rather than a gradual expansion into the Laurentide-Eurasian configuration, as is often assumed". Then, the growth of BerIS leads to the ice sheet-climate feedbacks (warming in the North Pacific and cooling over North America), which melt the BerIS itself and favor the enlargement of the Laurentide ice sheet.

In summary, our current study provides solid evidence to reconsider the BerIS. The mainstream Laurentide-Eurasia-only concept must reconcile these climate records summarized in our current study to strengthen itself.